# LipoPU: Pocket-level Prediction of Lipid-Protein Interactions via Positive-Unlabeled Learning

**Yuxing Wang** [1 2]  **Wenyi Zhang** [1 2]  **Yilong Zou** [1 2]  **Jing Huang** [† 1 2]

## Abstract

Computational identification of lipid-binding proteins is critical for both fundamental research and therapeutic development. Existing models are typically trained in a fully supervised manner, treating unlabeled samples as negatives. However, missing evidence does not imply non-binding, leading to systematic false negatives. Pocket-level lipid-binding prediction also remains underexplored compared to residue- or sequence-level approaches. To bridge these gaps, we present **LipoPU**, a pocket-centric predictor that formulates lipid-binding learning under a ranking-based positive-unlabeled objective, and supports both binary lipid-binding detection and multi-label lipid category prediction. LipoPU learns an attention-based pocket representation that is robust to ambiguous pocket definitions while providing residue-level interpretability. Experiments show consistent gains over supervised baselines and prior pocket-level work, and a structural case study recovers a literature-supported allosteric lipid-binding pocket while highlighting biologically informative residues. The code is released at https://github.com/JingHuangLab/LipoPU.

## 1. Introduction

Lipid-protein binding regulates diverse cellular processes, including signaling, transport, and metabolism, and has been implicated in a wide range of diseases (Saliba et al., 2015; Thiam et al., 2013; Harayama & Riezman, 2018). Motivated by this, accurate identification of lipid-binding proteins is

essential for both fundamental research and therapeutic development.

Recent advances in data-driven and deep learning models have accelerated computational lipid-binding prediction. However, training typically relies on experimentally curated annotations that provide only partial coverage: the absence of experimental evidence often reflects gaps in current knowledge, rather than true non-binding (Yu et al., 2014; Youngs et al., 2014). Many proteins that are currently unlabeled may be confirmed as lipid-binding as new structural or biochemical evidence emerges. A notable example is the GPR119 protein, whose orthosteric pocket within the seven transmembrane helices was long thought to accommodate only exogenous small-molecule agonists; only recently did high-resolution cryo-EM structures unexpectedly reveal an endogenous lysophosphatidylcholine (LPC) molecule occupying the same pocket, demonstrating specific lipid-binding capability (Xu et al., 2022). Such findings suggest that existing annotations represent a lower bound, and the landscape of lipid binding likely extends well beyond current understanding. Yet most current predictors do not explicitly model this annotation gap and instead adopt a fully supervised paradigm, treating annotated instances as positives and all others as negatives. This practice can introduce large numbers of false negatives and induce systematic decision bias, thereby hindering generalization and undermining the ability of computational methods to discover previously uncharacterized interactions at scale.

Beyond label incompleteness, progress is further constrained by the choice of prediction granularity. Most existing methods formulate lipid-binding prediction at the level of whole protein sequences (Dong & Wu, 2025; González-Díaz et al., 2012) or individual residues (Katuwawala et al., 2022; Malebary & Alromema, 2024). Yet lipid binding is typically mediated by localized structural sites that create specific geometric and physicochemical environments (Muller et al., 2019; Corradi et al., 2019; Bukiya & Dopico, 2017). These localized sites therefore constitute a natural biological decision unit for assessing lipid-binding capability. Modeling lipid binding at such sites, commonly referred to as pockets, offers a practical mechanistic abstraction. Despite its relevance, pocket-level lipid-binding

---

[1]State Key Laboratory of Gene Expression, School of Life Sciences, Westlake University, Hangzhou, Zhejiang, China [2]Westlake Laboratory of Life Sciences and Biomedicine, Hangzhou, Zhejiang, China. Correspondence to: Jing Huang <huangjing@westlake.edu.cn>.

*Proceedings of the 43rd International Conference on Machine Learning*, Seoul, South Korea. PMLR 306, 2026. Copyright 2026 by the author(s).

prediction remains underexplored and is inherently challenging. Unlike residues or full-length proteins, pockets lack a canonical definition and are usually inferred through heuristic detection algorithms, leading to variability in pocket boundaries that can complicate model training.

Another largely underexplored challenge is lipid specificity. While binary prediction asks whether a protein binds any lipids at all, lipid-protein interactions are frequently category-specific (Harayama & Riezman, 2018; Muller et al., 2019; Corradi et al., 2019). For instance, phosphoinositide-protein interactions regulate cell survival and metabolism (Ebner et al., 2017), whereas sterol-protein interactions play essential roles in embryogenesis and cancer (Huang et al., 2016). To organize lipid chemical diversity, LIPID MAPS (Sud et al., 2007; Conroy et al., 2024) defines eight major lipid categories: *Fatty Acyls* (FA), *Glycerolipids* (GL), *Glycerophospholipids* (GP), *Sphingolipids* (SP), *Sterol Lipids* (ST), *Prenol Lipids* (PR), *Saccharolipids* (SL), and *Polyketides* (PK). Due to structural complexity, many lipids can be associated with multiple categories. This diversity in lipid chemistry makes category-level prediction more informative than a binary label, yet most existing methods remain restricted to binary lipid-binding detection and provide limited insight into lipid category specificity.

To address these limitations, we introduce **LipoPU**, a pocket-level lipid-protein binding predictor formulated under a ranking-based positive-unlabeled (PU) learning setting. LipoPU learns a pocket's latent lipid-binding propensity, treating pockets without lipid-bound evidence as unlabeled rather than negative. Beyond binary binding detection, LipoPU further models pocket-lipid associations in a multi-label setting to identify the specific lipid categories involved. For each candidate pocket, LipoPU uses attention-based MIL pooling to aggregate residue-level embeddings into pocket-level representations. This provides a flexible pooling mechanism under ambiguous pocket boundaries and supports residue-level interpretation through learned attention weights. Training is guided by a ranking-based PU objective together with a tailored scheme designed for highly imbalanced data. Extensive evaluations demonstrate that LipoPU consistently outperforms supervised baselines and the existing pocket-level method, and a case study further highlights its ability to recover biologically meaningful lipid-binding pockets in realistic settings. In summary, the principal contributions of this study are as follows:

- We introduce **LipoPU**, a systematic framework for pocket-level lipid-protein binding prediction that explicitly models incomplete binding annotations through a ranking-based PU learning paradigm and aligns the prediction unit with pockets as natural biological decision units;

- We formulate two coupled prediction problems: (i) bi-

nary lipid-binding detection, which identifies whether a pocket exhibits lipid-binding capability, and (ii) multi-label lipid category prediction, which estimates the pocket's binding propensities for specific lipid categories;

- We employ a learnable attention-based pocket representation that accommodates potentially ambiguous pocket boundaries and enables residue-level interpretation through the learned attention distribution over pocket residues.

## 2. Related Works

### 2.1. Modeling Lipid-Protein Binding

Computational approaches provide a scalable complement to experimental studies for characterizing lipid-protein binding. While molecular dynamics (MD) simulations are widely used to study lipid-binding mechanisms at atomic resolution, their computational cost limits large-scale analysis (Muller et al., 2019). To address scalability, recent work has increasingly explored data-driven methods based on molecular representations and machine learning.

Existing learning-based approaches can be broadly organized by the granularity at which lipid binding is modeled. At the residue level, prior work has largely focused on intrinsically disordered regions (IDRs). Methods such as DisoLipPred (Katuwawala et al., 2022) and iDLB-Pred (Malebary & Alromema, 2024) apply deep learning to identify lipid-binding residues within IDRs, while DisoFLAG (Pang & Liu, 2024) leverages protein language models to detect IDRs and their associated functions, including lipid binding. These methods capture local residue signals but do not explicitly model spatially coherent binding sites.

Protein pockets are key structural and functional units for molecular recognition, yet pocket-level lipid-binding prediction remains underexplored. To the best of our knowledge, SLiPP (Chou et al., 2025; Chou & Dassama, 2024) is the only method that explicitly targets lipid-binding pockets. SLiPP operates on fpocket (Le Guilloux et al., 2009; Schmidtke et al., 2010)-predicted cavities, rather than experimentally annotated binding sites, and represents each pocket using a set of 17 handcrafted physicochemical descriptors defined in fpocket. It then applies classical machine learning models (e.g., random forests) for supervised lipid-binding prediction; additional details are provided in the Appendix D.1.

Most existing approaches formulate lipid binding as a binary prediction task, addressing whether binding occurs but not which lipid categories are involved. Very recently, PLi-Cat (Dong & Wu, 2025) extends lipid-binding prediction to the category level using full-length protein representations

derived from protein language models. However, sequence-level prediction cannot localize which residues collectively form the binding site, a limitation that is particularly restrictive for lipid binding, where pockets may be locally conserved despite substantial sequence variation (Sandhu et al., 2018).

Across these granularities, existing methods predominantly adopt fully supervised training paradigms that treat unlabeled instances as negatives, without explicitly modeling annotation incompleteness. Moreover, these methods provide limited insight into how residue groups and structural elements contribute to lipid-binding predictions, thereby constraining biological interpretability. Together, these limitations motivate the positive-unlabeled formulation and pocket-centric modeling strategy introduced in this work.

## 2.2. PU Learning for Biological Applications

Positive-unlabeled learning addresses classification settings in which only positive labels are reliable, while unlabeled data may contain latent positives (Du Plessis et al., 2015). This formulation naturally aligns with biological prediction tasks, where experimental annotations are sparse, incomplete, and biased toward well-studied entities.

To date, PU learning has been widely adopted across bioinformatics and has demonstrated strong effectiveness under incomplete labeling (Li et al., 2022). In disease gene prediction, methods such as NIAPU (Stolfi et al., 2023) cast the identification of novel disease-associated genes as a PU learning problem, enabling discovery beyond curated positives. In the context of Gene Ontology (GO) annotation, PU-GO (Zhapa-Camacho et al., 2024) leverages ESM-2 (Lin et al., 2023) protein language model representations and adopts a ranking-based PU loss (Tang et al., 2022) to encourage positive samples to be ranked above unlabeled ones, while incorporating ontology-derived priors as class-level label priors for PU training. Beyond these areas, PU learning has also been explored in a range of other biological applications, including pupylation site prediction (Jiang & Cao, 2016; Nan et al., 2017), gene regulatory network inference (Patel & Wang, 2015), and protein-RNA interaction prediction (Cheng et al., 2015), highlighting its ability to reduce false-negative bias and improve robustness to annotation incompleteness.

Despite this progress, PU learning has not yet been systematically explored for lipid-protein interaction prediction, particularly at the level of structurally localized binding pockets and across diverse lipid categories, motivating the formulation introduced in this work.

## 3. Method

### 3.1. LipoPU Overview

Given a query pocket, LipoPU addresses two closely related tasks: lipid-binding detection, which determines whether the pocket can bind any lipid, and lipid category prediction, which outputs multi-label, category-specific binding scores. Figure 1 illustrates the overall architecture of LipoPU.

Starting from a full-length protein sequence annotated with a pocket, LipoPU first encodes the sequence using a frozen ESM-2 (Lin et al., 2023) pretrained model with 650M parameters, producing a 1280-dimensional embedding for each residue. We then apply pocket residue selection by gathering embeddings at the annotated pocket positions. These preprocessing steps are training-free and do not receive gradients, remaining fixed during optimization.

LipoPU next constructs a pocket-level representation using attention-based multiple instance learning (MIL) pooling, which aggregates selected residue embeddings with learned attention weights. The pocket representation is then fed into a lightweight multi-layer perceptron (MLP) classifier to produce binding scores for each lipid category. The pooling module and the MLP classifier are trained jointly under a ranking-based PU objective. Lipid-binding detection is derived from the category scores by taking their maximum, linking the two tasks through a shared predictor.

Training is paired with a PU-specific batch construction strategy, centered on a positive-guaranteed batch sampler that enforces sufficient positive samples in each batch. This design stabilizes ranking-based PU optimization under severe class imbalance.

The rest of this section is organized as follows. Section 3.2 describes the pocket-aware MIL pooling module, Section 3.3 presents the ranking-based multi-label PU objective, and Section 3.4 details the PU batch construction procedure.

### 3.2. Pocket-aware MIL Pooling

We adopt an attention-based multiple instance learning formulation for pocket-level representation, following prior work that parameterizes the Bernoulli bag label probability with neural networks (Ilse et al., 2018).

A protein pocket is represented as a set of $M$ residue embeddings $X = \{x_1, \ldots, x_M\}$, where each $x_k \in \mathbb{R}^D$ corresponds to a residue within the pocket. To enable residue-wise weighting while preserving permutation invariance over the set, each residue embedding is first projected into a latent attention space:

$$h_k = \tanh(V x_k), \tag{1}$$

where $V \in \mathbb{R}^{H \times D}$ is a learnable projection matrix and $H$

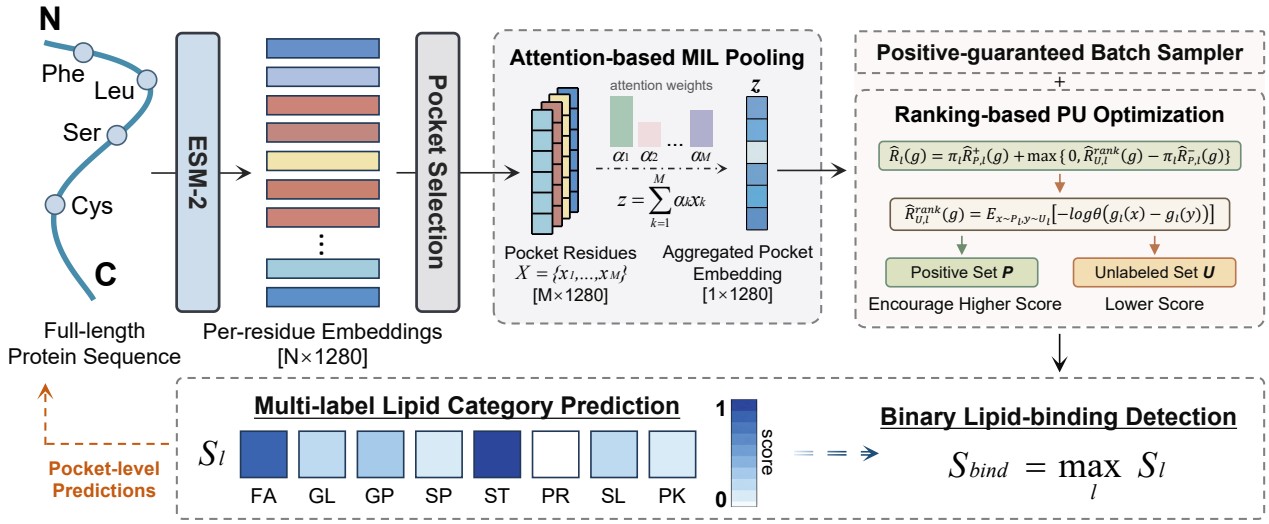

*Figure 1.* Overview of the LipoPU framework. A frozen ESM-2 encoder extracts residue-level embeddings from the full-length protein sequence, after which residues belonging to candidate pockets are selected. The selected residues are aggregated into a fixed-dimensional pocket representation via attention-based MIL pooling. A positive-guaranteed batch sampler and a ranking-based PU objective are used to train a multi-label classifier that outputs lipid category scores $S_l$. Binary lipid-binding detection is obtained by $S_{\text{bind}} = \max_l S_l$.

denotes the attention hidden dimension.

Attention weights are then computed as:

$$\alpha_k = \frac{\exp(w^\top h_k)}{\sum_{j=1}^{M} \exp(w^\top h_j)}, \quad (2)$$

where $w \in \mathbb{R}^H$ is a learnable attention vector.

Finally, the pocket-level representation is obtained via weighted aggregation:

$$z = \sum_{k=1}^{M} \alpha_k x_k, \quad z \in \mathbb{R}^D. \quad (3)$$

This formulation assigns different importance weights to individual residues, while remaining invariant to the ordering of residues within a pocket.

Such attention-based MIL pooling provides a flexible mechanism for representing protein pockets, whose boundaries are often ambiguous and detector-dependent. Moreover, because residues within a pocket may contribute unequally to lipid binding, the learned attention weights serve as a useful proxy for residue-level interpretation by highlighting residues that are more informative for pocket-level lipid-binding predictions.

### 3.3. PU Learning Objective for Lipids

The aggregated pocket representations obtained by attention-based MIL pooling are fed into an MLP classifier, which outputs binding scores for each lipid category. Our goal is not only to assign prediction scores but also to encourage

the prioritization of experimentally confirmed lipid-binding pockets in the ranking, while acknowledging that the unlabeled set may contain latent positives. To achieve this, we adopt a ranking-based PU objective for multi-label lipid-binding prediction, obtained by reformulating the standard supervised binary classification risk.

In the fully supervised binary classification setting, the risk of a decision function $g(\cdot) : \mathbb{R}^D \to \mathbb{R}$ is defined as:

$$\widehat{R}(g) = \pi \widehat{R}_P^+(g) + (1 - \pi)\widehat{R}_N^-(g), \quad (4)$$

where $\pi = \mathbb{P}(y = 1)$ denotes the positive class prior, $\widehat{R}_P^+(g)$ denotes the risk on positive samples when they are treated as positives, and $\widehat{R}_N^-(g)$ denotes the risk on negative samples when they are treated as negatives.

Under the positive-unlabeled setting, negative samples are not directly observable. Instead, unlabeled samples form a mixture distribution $U$ that contains both true negatives and unannotated positives. Following prior work (Du Plessis et al., 2015), the negative risk term can be rewritten by subtracting the positive-contamination term from the unlabeled risk, which provides a key correction for false-negative bias:

$$(1 - \pi)\widehat{R}_N^-(g) = \widehat{R}_U^-(g) - \pi \widehat{R}_P^-(g), \quad (5)$$

where $\widehat{R}_U^-(g)$ denotes the risk of treating unlabeled samples as negatives, and $\widehat{R}_P^-(g)$ denotes the risk of misclassifying positive samples as negatives. Substituting Eq. (5) into Eq. (4) yields the standard PU risk formulation:

$$\widehat{R}(g) = \pi \widehat{R}_P^+(g) + \widehat{R}_U^-(g) - \pi \widehat{R}_P^-(g). \quad (6)$$

To ensure that the empirical risk remains non-negative during optimization, we follow the non-negative PU risk estimator (Kiryo et al., 2017) and extend it to the multi-label prediction setting. For lipid-binding category $l$, the risk is defined as:

$$\widehat{R}_l(g) = \pi_l \widehat{R}_{P,l}^+(g) + \max\{0, \widehat{R}_{U,l}^-(g) - \pi_l \widehat{R}_{P,l}^-(g)\}, \quad (7)$$

where $P_l$ and $U_l$ denote the observed positive and unlabeled sets for this category, respectively, and $\pi_l$ denotes its positive prior.

In standard PU learning, the risk terms are typically estimated using pointwise logistic losses. For a sample $x \in \mathbb{R}^D$, the corresponding risks for label $l$ are defined as:

$$\widehat{R}_{P,l}^+(g) = \mathbb{E}_{x \sim P_l}\left[-\log \theta(g_l(x))\right], \quad (8)$$

$$\widehat{R}_{P,l}^-(g) = \mathbb{E}_{x \sim P_l}\left[-\log \theta(-g_l(x))\right], \quad (9)$$

$$\widehat{R}_{U,l}^-(g) = \mathbb{E}_{x \sim U_l}\left[-\log \theta(-g_l(x))\right], \quad (10)$$

where $\theta(\cdot)$ denotes the sigmoid function, and $g_l(\cdot)$ is the decision function for lipid category $l$. Intuitively, $\widehat{R}_{P,l}^+(g)$ encourages observed positives to receive high scores, $\widehat{R}_{P,l}^-(g)$ measures the penalty that would arise if observed positives were treated as negatives, and $\widehat{R}_{U,l}^-(g)$ penalizes high scores assigned to unlabeled samples when they are treated as negatives.

In practice, relative ordering is often more informative than absolute scores. Following prior work on PU ranking objectives (Zhapa-Camacho et al., 2024; Tang et al., 2022), we replace the unlabeled-negative term $\widehat{R}_{U,l}^-(g)$ with a ranking-based PU surrogate $\widehat{R}_{U,l}^{\text{rank}}(g)$, thereby converting the absolute low-score constraint on unlabeled pockets into a relative ordering constraint between observed positives and unlabeled pockets:

$$\widehat{R}_{U,l}^{\text{rank}}(g) = \mathbb{E}_{x \sim P_l, \, y \sim U_l}\left[-\log \theta\big(g_l(x) - g_l(y)\big)\right]. \quad (11)$$

With this ranking surrogate, we define the category-wise PU objective used in LipoPU as:

$$\widehat{R}_l(g) = \pi_l \widehat{R}_{P,l}^+(g) + \max\{0, \widehat{R}_{U,l}^{\text{rank}}(g) - \pi_l \widehat{R}_{P,l}^-(g)\}. \quad (12)$$

Finally, the overall objective of LipoPU is obtained by summing the risks across all lipid categories:

$$\mathcal{L}(g) = \sum_{l=1}^{L} \widehat{R}_l(g), \quad (13)$$

where $L$ denotes the total number of lipid categories.

---

**Algorithm 1** Positive-guaranteed Batch Sampler
---
**Input:** dataset of size $N$, positive set $P$, unlabeled set $U$, batch size $B$, min_pos, max_pos, $\rho$
// Compute number of batches
$T = \lceil N/B \rceil$
// Compute positive counts per batch
Initialize pos_count$[1..T] \approx |P|/T$
Add noise to each pos_count$[t]$ controlled by $\rho$
Clip each pos_count$[t]$ to [min_pos, max_pos]
Randomly shuffle $P$ and $U$
// Build each batch
**for** $t = 1$ **to** $T$ **do**
    pick pos_count$[t]$ positives from $P$ into batch $t$
    **if** $P$ runs out **then**
        reshuffle $P$ and continue picking
    **end if**
    pick $(B - $pos_count$[t])$ unlabeled samples from $U$ into batch $t$
    **if** $U$ runs out **then**
        randomly sample remaining indices from all $\{0, \dots, N-1\}$
    **end if**
    output batch $t$
**end for**

---

### 3.4. Batch Construction for PU Learning

Our PU objective relies on within-batch ranking between positive and unlabeled samples. When a training batch contains only unlabeled instances, the objective provides no learning signal due to the absence of positive-unlabeled comparisons. To stabilize ranking-based PU training under severe class imbalance, we introduce a positive-guaranteed batch sampler that enforces each mini-batch to contain at least a preset minimum number of positives (Algorithm 1).

Here, $N$ is the number of training instances. $P$ and $U$ denote sets of positives and unlabeled samples, and $B$ is the mini-batch size. min_pos and max_pos control the lower and upper bounds on positives per batch, while $\rho$ controls the amount of randomness in the positive counts. In practice, this sampler is tightly coupled with the PU objective, keeping each mini-batch informative for the ranking loss.

## 4. Experiments

We conduct extensive experiments to systematically evaluate LipoPU. The experimental setup is described in Section 4.1. Our experiments are designed to address the following four questions. **Q1:** Does ranking-based positive-unlabeled learning provide a more appropriate learning paradigm than absolute labeling for pocket-level lipid-binding prediction? **Q2**: Can LipoPU improve upon prior pocket-level work? **Q3**: How does each component of LipoPU contribute to

overall performance? **Q4**: Can LipoPU make reliable predictions on realistic pocket instances and provide interpretable signals at the residue level?

## 4.1. Experimental Setup

**Datasets.** Our pocket-level dataset is constructed from BioLiP2 (Zhang et al., 2024), a curated resource of protein-ligand interactions with annotated binding sites. Each annotated site is treated as a pocket instance, without enforcing a strict cavity geometry, as many binding sites correspond to shallow or surface-exposed regions. Lipid-binding annotations are referenced from BioDolphin (Yang et al., 2024). While LipoPU aims to model a pocket's latent lipid-binding propensity, a pocket observed with a non-lipid ligand in BioLiP2 may still accommodate lipids under different contexts. Therefore, BioLiP2 pockets interacting with lipid molecules listed in BioDolphin are labeled as positive, while all remaining pockets are treated as unlabeled.

After systematic curation and filtering, the final dataset contains 395,959 pockets, including 18,402 positive samples. The dataset exhibits substantial imbalance between positive and unlabeled pockets, as well as across lipid categories. As shown in Figure 4, individual positive pockets may be associated with multiple lipid categories. Protein sequences are clustered using MMseqs2 (Steinegger & Söding, 2017) at a 30% identity threshold and split into training, validation, easy test, and hard test sets (65% / 5% / 20% / 10%). The hard test set shares less than 30% sequence identity with all other splits, representing a challenging remote-homology evaluation, while the easy test set reflects a realistic near-distribution setting. This split design enables us to separately assess generalization under both near- and remote-homology conditions. Full preprocessing details and split statistics are provided in Appendix A.

**Evaluation tasks and baselines.** Our evaluation includes two tasks: (i) **lipid-binding detection**, which assesses a model's ability to identify lipid-binding pockets, and (ii) **lipid category prediction**, which evaluates its ability to characterize lipid specificity. Results are reported on both easy and hard test sets. We first compare LipoPU against supervised baselines, where unlabeled pockets are treated as negatives. These baselines correspond to conventional absolute-label training under incomplete annotation. We further compare with SLiPP (Chou et al., 2025; Chou & Dassama, 2024), the only existing method that performs lipid-binding prediction at the pocket level. Implementation and training details are provided in Appendix B.

**Evaluation Metrics.** We evaluate models using ROC-AUC and PR-AUC, two threshold-free metrics that quantify ranking quality. ROC-AUC reflects the model's ability to rank positives ahead of others, while PR-AUC emphasizes

precision-recall trade-offs for the positive class and is more informative under severe class imbalance. For lipid-binding detection, we report Binary ROC-AUC and Binary PR-AUC, which assess whether a pocket binds any lipid regardless of category. For lipid category prediction, we report Micro ROC-AUC and Micro PR-AUC, which measure overall ranking performance across all lipid categories. We also report Per-category ROC-AUC and Per-category PR-AUC for individual lipid classes. To further evaluate the model's ability to prioritize lipid-binding pockets, we additionally report Top-$k$ precision for lipid-binding detection. Details of metrics are provided in Appendix C.

## 4.2. Effect of PU Learning (Q1)

We compare LipoPU with supervised baselines trained using absolute labels, where pockets without experimental evidence of lipid interactions are treated as negatives. These baselines are optimized using weighted binary cross-entropy (BCE). Because batch construction can influence training under extreme class imbalance, we explicitly construct two BCE baselines: (i) **BCE (no sampler)**, trained without batch-level constraints, and (ii) **BCE (sampler)**, trained with our positive-guaranteed batch sampler that enforces a minimum number of positive samples in each batch. All methods share the same backbone architecture, pocket representation, and evaluation protocol.

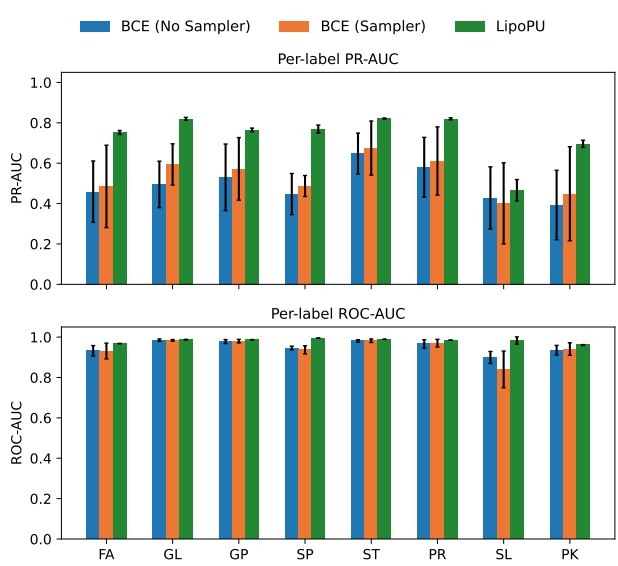

*Figure 2.* Per-label performance on the Easy test set. Bars report PR-AUC (top) and ROC-AUC (bottom) for each lipid category, comparing fully supervised BCE baselines (with/without the positive-guaranteed batch sampler) against LipoPU. LipoPU improves consistently across categories, with the largest gains in PR-AUC; error bars denote variability across runs.

*Table 1.* Supervised BCE vs. LipoPU on lipid-binding detection (Binary) and lipid category prediction (Micro). Metrics are reported separately on the easy and hard test sets. Results are reported as mean ± std. Best results are shown in **bold**.

| Set | Model | ROC-AUC (↑) | | PR-AUC (↑) | |
|-----|-------|-------------|--------------|------------|--------------|
| | | Binary | Micro | Binary | Micro |
| Easy | BCE (no sampler) | $0.942 \pm 0.025$ | $0.972 \pm 0.011$ | $0.630 \pm 0.118$ | $0.453 \pm 0.128$ |
| | BCE (sampler) | $0.938 \pm 0.041$ | $0.972 \pm 0.014$ | $0.651 \pm 0.154$ | $0.490 \pm 0.155$ |
| | **LipoPU** | **$0.973 \pm 0.001$** | **$0.984 \pm 0.0004$** | **$0.839 \pm 0.009$** | **$0.766 \pm 0.010$** |
| Hard | BCE (no sampler) | $0.908 \pm 0.011$ | $0.906 \pm 0.011$ | $0.557 \pm 0.061$ | $0.195 \pm 0.027$ |
| | BCE (sampler) | $0.904 \pm 0.019$ | $0.898 \pm 0.014$ | $0.558 \pm 0.069$ | $0.199 \pm 0.033$ |
| | **LipoPU** | **$0.914 \pm 0.003$** | **$0.924 \pm 0.007$** | **$0.650 \pm 0.013$** | **$0.280 \pm 0.015$** |

*Table 2.* SLiPP vs. LipoPU on lipid-binding detection (Binary). Metrics are reported separately on the easy and hard fpocket-derived test sets. Results are reported as mean ± std. LipoPU additionally enables lipid category prediction (Micro). Best results are shown in **bold**.

| Set | Model | ROC-AUC (↑) | | PR-AUC (↑) | |
|-----|-------|-------------|--------------|------------|--------------|
| | | Binary | Micro | Binary | Micro |
| Easy-fpocket | SLiPP | $0.742 \pm 0.001$ | – | $0.239 \pm 0.002$ | – |
| | **LipoPU** | **$0.960 \pm 0.003$** | **$0.985 \pm 0.001$** | **$0.792 \pm 0.013$** | **$0.727 \pm 0.012$** |
| Hard-fpocket | SLiPP | $0.726 \pm 0.003$ | – | $0.264 \pm 0.004$ | – |
| | **LipoPU** | **$0.881 \pm 0.005$** | **$0.907 \pm 0.004$** | **$0.505 \pm 0.018$** | **$0.209 \pm 0.021$** |

Table 1 provides results for LipoPU and supervised BCE baselines on lipid-binding detection and lipid category prediction, with binary and micro-averaged metrics. LipoPU consistently outperforms supervised baselines across both tasks on both easy and hard test sets. Introducing a positive-guaranteed batch sampler leads to only minor changes in supervised BCE performance, indicating that the observed gains arise primarily from the PU learning objective rather than batch sampling. In particular, LipoPU achieves substantial improvements in PR-AUC, highlighting its effectiveness in identifying lipid-binding pockets under severe class imbalance.

Figure 2 further presents per-label results on the easy test set, where each label corresponds to a lipid category. LipoPU outperforms supervised baselines across all lipid categories, demonstrating that its advantage is consistent and not driven by a small subset of dominant categories. Nevertheless, category-specific scores should be interpreted with additional caution for extremely rare lipid classes, especially SL and SP, since these categories contain fewer observed positives for training and yield less stable estimates during evaluation. On the hard test set, performance decreases for all methods as expected, reflecting the increased difficulty of lipid-binding prediction under remote homology. Despite this challenge, LipoPU remains consistently stronger than supervised baselines. In addition, LipoPU exhibits superior ranking performance on lipid-binding detection, achieving over 0.98 precision within the Top-100 predictions on both test sets, indicating effective prioritization of lipid-binding pockets. Detailed per-label results on the hard test set and ranking analyses are reported in Appendix E.1 and E.2. Taken together, these results indicate that PU learning provides a more effective learning paradigm than absolute labeling in the presence of incomplete and ambiguous pocket annotations.

### 4.3. Comparison with SLiPP (Q2)

To the best of our knowledge, SLiPP (Chou et al., 2025; Chou & Dassama, 2024) is among the very few existing methods that perform lipid-binding prediction explicitly at the pocket level. It adopts a supervised pipeline based on fpocket-derived pocket definitions and handcrafted descriptors, which differs substantially from LipoPU in both representation and learning formulation.

To ensure a fair comparison under a unified pocket definition, we construct fpocket-based test sets for both easy and hard splits. Specifically, fpocket is applied to the experimental protein structures corresponding to test-set pockets, and predicted pockets are then matched to annotated binding sites based on maximum residue overlap. Pockets with sufficient overlap are assigned the corresponding labels, yielding **easy-fpocket** and **hard-fpocket** test sets. Detailed construction procedures are provided in the Appendix D.2. During evaluation, SLiPP operates on its original fpocket descriptors, whereas LipoPU derives ESM-2 representations from the same fpocket-predicted pocket residues.

As shown in Table 2, LipoPU consistently outperforms SLiPP on lipid-binding detection across both fpocket-based test sets. Improvements are especially evident in PR-AUC, where LipoPU achieves approximately 2–3× higher precision-recall performance, reflecting more effective prioritization of lipid-binding pockets. Beyond binary lipid-binding detection, LipoPU further enables lipid category pre-

diction, a capability not supported by SLiPP, and achieves competitive micro-averaged performance, with per-label results shown in Appendix E.4. Notably, these evaluations are conducted on fpocket-derived pockets rather than the experimentally annotated pockets used for LipoPU training. Despite this mismatch, LipoPU maintains robust performance, demonstrating effective generalization across different pocket definitions.

## 4.4. Ablation Study (Q3)

Table 3 reports an ablation study that examines the contribution of individual components in LipoPU. We first analyze the batch construction strategy by removing the positive-guaranteed batch sampler. This variant shows noticeable performance drops, indicating that enforcing a minimum number of positive samples per batch is critical for stable optimization under extreme class imbalance. Next, we replace the proposed ranking-based PU loss with a standard PU formulation. This change results in substantial degradation across all metrics. Notably, the decline is most pronounced in PR-AUC, highlighting the importance of explicitly modeling relative ranking in PU learning, particularly for lipid category prediction. Finally, we compare attention-based MIL pooling with mean pooling. Mean pooling remains a strong aggregation baseline and achieves competitive performance on several binary metrics. Nevertheless, attention-based MIL pooling shows modest advantages for lipid category prediction, especially in terms of PR-AUC. The per-category analysis in Appendix E.5 shows that the attention-based variant achieves higher PR-AUC than mean pooling for most lipid classes, suggesting that learnable residue weighting may help capture category-discriminative signals at the pocket level. Beyond these performance differences, a key practical advantage of attention-based MIL pooling is that it provides learned residue-level attention weights, which support residue-level interpretation by indicating which residues contribute more strongly to the pocket-level lipid-binding prediction.

## 4.5. Case Study (Q4)

The human angiotensin II type 1 receptor (AT1R; PDB ID: 7F6G) is a key therapeutic target in cardiovascular disease. It ranks among the top 20 predictions in our hard test set ($<30\%$ sequence identity to training data). Prior work (Lu et al., 2021) identified a cryptic allosteric pocket in an intermediate activation state of AT1R using MD simulations and fpocket (Le Guilloux et al., 2009; Schmidtke et al., 2010). Subsequent cryo-EM structures directly resolved a cholesterol molecule bound within an intracellular cavity formed by TM1, TM7, and helix 8 (Zhang et al., 2023), as shown in Figure 3. Functional assays further demonstrated that residues forming this pocket (e.g., $F^{1.48}$, $F^{7.55}$, $K^{8.51}$, $F^{8.54}$) modulate both G protein and $\beta$-arrestin signaling.

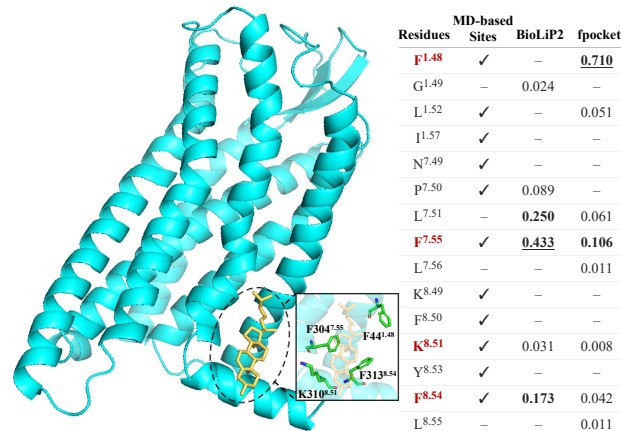

| Residues | MD-based Sites | BioLiP2 | fpocket |
|---|---|---|---|
| $F^{1.48}$ | ✓ | – | 0.710 |
| $G^{1.49}$ | – | 0.024 | – |
| $L^{1.52}$ | ✓ | – | 0.051 |
| $I^{1.57}$ | ✓ | – | – |
| $N^{7.49}$ | ✓ | – | – |
| $P^{7.50}$ | ✓ | 0.089 | – |
| $L^{7.51}$ | – | **0.250** | 0.061 |
| $F^{7.55}$ | ✓ | **0.433** | **0.106** |
| $L^{7.56}$ | – | – | 0.011 |
| $K^{8.49}$ | ✓ | – | – |
| $F^{8.50}$ | ✓ | – | – |
| $K^{8.51}$ | ✓ | 0.031 | 0.008 |
| $Y^{8.53}$ | ✓ | – | – |
| $F^{8.54}$ | ✓ | **0.173** | 0.042 |
| $L^{8.55}$ | – | – | 0.011 |

*Figure 3.* Case study on AT1R (PDB 7F6G). **Left**: cryo-EM structure with a cholesterol molecule bound in an intracellular cavity; the inset zooms into the site and highlights key residues. **Right**: residue-level attention assigned by LipoPU for the BioLiP2-annotated and fpocket-derived pockets, shown alongside MD-based sites from prior work. In the table, ✓/– indicate whether a residue is included by each source, and values report per-residue attention weight within the corresponding pocket; **bold** marks high-attention residues and underlining denotes the maximum. LipoPU predicts sterol-lipid binding with near-unity confidence and concentrates attention on literature-supported pocket residues.

To align with prior experimental setups (Lu et al., 2021), we consider both BioLiP2-annotated and fpocket-derived pockets from the cryo-EM structure. LipoPU assigns near-unity prediction scores under both pocket definitions, scoring 0.999 for the BioLiP2 annotation and 0.987 for the fpocket-derived pocket, and correctly identifies sterol lipids as the top-ranked binding category (Appendix E.6). As shown in Figure 3, over 80% of the attention mass is concentrated on residues $L^{7.51}$, $F^{7.55}$, and $F^{8.54}$ in the BioLiP2 pocket. These residues define a hydrophobic cavity between TM7 and H8 and have been linked to allosteric modulation of both G protein and $\beta$-arrestin signaling (Lu et al., 2021). For the fpocket-derived pocket, LipoPU assigns 71% of the attention to $F^{1.48}$, a residue absent from the BioLiP2 annotation but previously reported to perturb receptor signaling (Lu et al., 2021).

This example highlights LipoPU's ability to identify functionally relevant lipid-binding allosteric pockets. The consistent high scores for both BioLiP2-annotated and fpocket-derived pockets suggest that the model can accommodate boundary variation in practical pocket-level prediction. Beyond the prediction scores, the attention weights provide a residue-level interpretation signal by highlighting residues assigned high importance in the pocket-level lipid-binding prediction. Although these weights should not be interpreted as direct mechanistic contributions to lipid binding, their

*Table 3.* Ablation study of sampling, loss design, and pooling strategy. Results are reported as mean ± std. All variants share the same backbone and differ only in the indicated component. Best results are shown in **bold**, and second-best results are underlined.

| Model | Easy Test Set | | | | Hard Test Set | | | |
|---|---|---|---|---|---|---|---|---|
| | ROC-AUC (↑) | | PR-AUC (↑) | | ROC-AUC (↑) | | PR-AUC (↑) | |
| | Binary | Micro | Binary | Micro | Binary | Micro | Binary | Micro |
| No Sampler | 0.966 ± 0.005 | 0.978 ± 0.002 | 0.799 ± 0.025 | 0.696 ± 0.033 | 0.905 ± 0.007 | 0.913 ± 0.004 | 0.644 ± 0.011 | 0.250 ± 0.005 |
| PU Loss | 0.899 ± 0.001 | 0.929 ± 0.0004 | 0.449 ± 0.002 | 0.261 ± 0.001 | 0.895 ± 0.001 | 0.926 ± 0.001 | 0.525 ± 0.003 | 0.208 ± 0.001 |
| Mean Pooling | **0.975 ± 0.001** | 0.983 ± 0.001 | 0.833 ± 0.007 | 0.731 ± 0.011 | **0.931 ± 0.003** | **0.928 ± 0.001** | **0.694 ± 0.007** | 0.269 ± 0.004 |
| LipoPU | 0.973 ± 0.001 | **0.984 ± 0.0004** | **0.839 ± 0.009** | **0.766 ± 0.010** | 0.914 ± 0.003 | 0.924 ± 0.007 | 0.650 ± 0.013 | **0.280 ± 0.015** |

agreement with experimentally implicated residues supports the biological relevance of the learned attention pattern.

## 5. Conclusion

In this study, we presented LipoPU, a pocket-centric predictor trained with a ranking-based positive-unlabeled objective. Across lipid-binding detection and lipid category prediction, LipoPU delivers consistent gains over fully supervised baselines that treat unlabeled pockets as negatives and prior pocket-level predictors, and it produces high-confidence rankings that support systematic prioritization at scale. Attention-based MIL pooling further provides learned residue-level attention weights, which support biological interpretation and hypothesis generation for candidate pockets with potentially ambiguous boundaries.

**Limitations remain.** Performance can degrade under substantial shifts in pocket definitions across data sources. Category-level prediction is constrained by shared domain patterns, limited positives for several lipid types, and coarse label granularity. Predictions for extremely rare lipid categories should therefore be interpreted with particular caution. LipoPU predictions should be interpreted as supporting ranking and candidate prioritization, rather than biological confirmation of binding. Furthermore, the annotation incompleteness that motivates PU learning can also affect both training and evaluation, since observed labels may be biased across lipid types. Despite these challenges, LipoPU provides a practical route to systematically screen unlabeled collections and to broaden our understanding of lipid-protein interactions beyond what is currently annotated.

## Acknowledgements

This work was supported by the Shandong Provincial Natural Science Foundation (ZR2024LZH009), the Shenzhen Medical Research Fund (B2404003), the National Natural Science Foundation of China (T2596084, 32501101), the State Key Laboratory of Gene Expression, and the Westlake Education Foundation. We thank the Westlake University Supercomputer Center for computational resources and related assistance.

## Impact Statement

LipoPU can positively impact biology and medicine by making lipid-protein interaction discovery more scalable and more localized. Since current annotations likely represent only a lower bound on lipid-protein binding, LipoPU enables large-scale screening of unlabeled proteins to broaden the lipid-protein interaction landscape. Its attention-based residue-level attributions provide interpretable signals that can guide mechanistic studies and support functional annotation and target discovery. As with any screening approach, predictions may be over-trusted and training data can be biased toward lipids that are easier to resolve, so LipoPU should be used to rank candidates rather than to confirm binding. When used this way and validated with experimental evidence when available, it offers a practical and responsible route to accelerate discovery.

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

# A. Datasets for LipoPU

## A.1. Data Sources and Pocket Definition

**BioLiP2 (structure-derived pockets).** BioLiP2 (Zhang et al., 2024) is a curated database of biologically relevant protein-ligand interactions. For each protein-ligand pair, BioLiP2 defines binding-site residues based on atomic contacts: all intermolecular distances between non-hydrogen atoms are computed, and an atom pair is considered in contact if their distance is within the sum of their van der Waals radii plus 0.5 Å. A residue is labeled as ligand-binding if it forms at least two such contacts to the ligand. Ligand-binding residues are then grouped into a binding site when the set contains at least two residues. Each BioLiP2 entry provides the protein chain sequence, the binding-site residue set, and the ligand identifier (PDB CCD code). However, BioLiP2 does not explicitly indicate whether a ligand is a lipid, nor does it provide lipid category labels.

**BioDolphin (lipid identity and categories).** BioDolphin (Yang et al., 2024) is curated specifically for lipid-protein interactions and provides lipid annotations following the LIPID MAPS classification system (8 lipid categories), along with functional and structural annotations. We use BioDolphin to obtain a comprehensive lipid CCD list for identifying lipid ligands and to assign lipid category labels to lipid-binding positives when available.

## A.2. Dataset preprocessing and curation

We curate pocket-ligand instances with a strict preprocessing pipeline to ensure structural reliability, sequence validity, and compatibility with pretrained ESM-2 (Lin et al., 2023) encoders. We apply the following filters:

- **Pocket validity.** Keep protein chains with sequence length $\geq 30$ residues and annotated binding-site size $\geq 3$ residues.

- **Structure reliability.** Retain only structures with experimental resolution $\leq 3$ Å.

- **Sequence consistency.** Discard entries where binding-site residues cannot be mapped consistently to the chain sequence (e.g., residue numbering mismatches), or where the sequence contains unknown residues (X).

- **Standard amino-acid alphabet.** Remove sequences containing non-standard or ambiguous residue types; keep only the 20 standard amino acids.

- **Redundancy removal.** Deduplicate identical tuples [protein sequence, binding-site residues, ligand CCD], primarily collapsing repeated occurrences from multi-chain complexes or repeated annotations.

- **ESM-2 length constraint.** Enforce sequence length $\leq 1024$ residues (ESM-2 input limit). For longer proteins, crop a binding-site-centered contiguous segment that contains the pocket while satisfying the 1024-token limit.

## A.3. Positive-Unlabeled Dataset Construction

Because the absence of experimental annotation does not imply *non*-lipid binding, we formulate lipid-pocket learning as a positive-unlabeled problem and construct the dataset as follows:

**Positive set (P).** Pocket instances whose ligand CCD appears in the BioDolphin lipid CCD list. Positives include BioLiP2 pockets whose ligands match the lipid CCD list, and additional BioDolphin instances with explicit binding-site residue annotations that are not present in BioLiP2.

**Unlabeled set (U).** BioLiP2 pocket instances whose ligand CCD does not appear in the BioDolphin lipid CCD list.

After preprocessing, the curated dataset contains 395,959 pocket instances, including 18,402 positives (~4.60%). Figure 4 summarizes the dataset statistics and the lipid-category distribution of positive pockets.

## A.4. Dataset Splitting

To assess both in-distribution performance and remote-homology generalization, we construct *easy* and *hard* test splits. We first cluster protein sequences using MMseqs2 (Steinegger & Söding, 2017) with a sequence identity threshold of 30% and a coverage threshold of 80%. We then hold out entire clusters until they account for ~10% of all instances, forming the hard test split. The remaining instances are further partitioned into training, validation, and easy test sets, where the easy split is intended to be more in-distribution relative to training. Detailed split sizes and per-lipid-category counts are reported in Table 4.

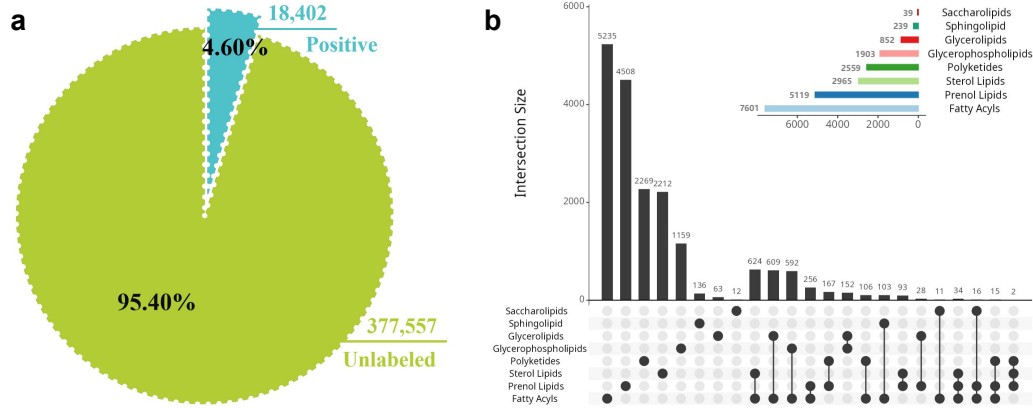

*Figure 4.* PU dataset construction and lipid-category composition. **a.** After preprocessing, our curated pocket dataset contains 395,959 instances, comprising 18,402 positives (4.60%) and 377,557 unlabeled pockets (95.40%), highlighting a severe class imbalance. **b.** Lipid-category distribution of the positive set according to the LIPID MAPS system (8 categories). The UpSet plot was generated using the online VennDetail web tool. The colored bar chart (upper right) shows the total number of pockets per category. The x-axis enumerates multi-label combinations; black dots mark the involved lipid categories, and vertical black bars report the number of pockets for each combination, indicating that positives may be associated with one or multiple lipid categories.

*Table 4.* Dataset split statistics and label distribution across lipid categories.

| Split | Size | Labeled | FA | GL | GP | SP | ST | PR | SL | PK |
|---|---|---|---|---|---|---|---|---|---|---|
| Train | 268,369 | 11,041 | 4,405 | 496 | 1,139 | 161 | 1,705 | 3,281 | 20 | 1,571 |
| Val | 17,558 | 1,714 | 755 | 6 | 143 | 19 | 192 | 557 | 4 | 235 |
| Easy test | 73,602 | 3,256 | 1,312 | 94 | 275 | 40 | 529 | 888 | 4 | 514 |
| Hard test | 36,430 | 2,391 | 1,129 | 256 | 346 | 19 | 539 | 393 | 11 | 239 |

## B. Implementation Details

We implement LipoPU in PyTorch (Paszke et al., 2019). We encode each protein with the pretrained ESM-2 (33-layer, 650M) model (Lin et al., 2023) and use the last-layer residue representations as input features. All ESM-2 parameters are frozen during training. Before MIL pooling, each pocket is represented as a variable-length set of residue embeddings. To enable efficient batching, we zero-pad residue sets within each batch to the maximum set size and apply a boolean mask so that padded tokens do not contribute to masked-attention MIL pooling.

The attention-based MIL pooling module uses a hidden dimension of 128. The multi-label classifier is a 3-layer MLP ($1280 \rightarrow 640 \rightarrow 320 \rightarrow 8$) with LayerNorm, GELU activations, and dropout 0.2. For PU learning, we estimate per-label class priors from training-set label frequencies, $\pi = [0.0328, 0.0037, 0.0085, 0.0012, 0.0127, 0.0245, 0.0010, 0.0117]$. In the PU ranking loss, we enforce at least 1 positive instance in each mini-batch, regardless of which label it belongs to. If a rare label has no observed instance in a mini-batch, positive supervision for that label is accumulated over the batches in which its positives appear. We control stochasticity in positive sampling via a randomness rate of 0.3 during training and 0 during validation.

In implementation, the risk terms in Sec. 3.3 are estimated by mini-batch averages using label-specific masks. For example, for a mini-batch $b$, let $P_l^{(b)}$ denote the observed positive subset for label $l$. The positive-dependent risks are calculated as:

$$\widehat{R}_{P,l}^{+}(g) = \frac{1}{|P_l^{(b)}|} \sum_{x \in P_l^{(b)}} -\log \theta(g_l(x)), \quad \widehat{R}_{P,l}^{-}(g) = \frac{1}{|P_l^{(b)}|} \sum_{x \in P_l^{(b)}} -\log \theta(-g_l(x)).$$

The unlabeled and ranking terms are estimated in the same mini-batch manner using the corresponding label-specific masks.

We train LipoPU with batch size 64 and learning rate 1e-4 using the Adam optimizer (Kingma & Ba, 2014) and weight decay 1e-4. We adapt the learning rate with a ReduceLROnPlateau schedule that halves the learning rate when the monitored validation criterion stops improving for 2 evaluations. All experiments are repeated 5 times with different random seeds on a single NVIDIA L40 GPU.

To mitigate overfitting, we employ early stopping with patience 5 and a maximum of 1000 epochs. We monitor validation

performance using a macro-averaged Average Precision (AP) criterion. Concretely, we apply a sigmoid to logits to obtain per-label probabilities and compute the macro-averaged AP against binary validation labels, defining

$$L_{\text{val}} = 1 - \text{AP},$$

where lower is better.

For weighted-BCE baselines, we use the same priors as LipoPU to derive class weights and the best-performing setting uses a learning rate of 1e-3. All other settings (batch size, optimizer, scheduler, early stopping, seeds, and hardware) are kept identical to LipoPU.

## C. Evaluation Metric Details

We evaluate models primarily with ROC-AUC and PR-AUC, two threshold-free metrics that assess ranking quality without committing to a specific decision threshold. ROC-AUC measures global ranking discrimination, reflecting the probability that a randomly chosen positively annotated instance is ranked above a randomly chosen other instance. PR-AUC emphasizes the positive class and is generally more informative under severe class imbalance.

We report the following metrics:

- **Binary ROC-AUC and Binary PR-AUC** evaluate lipid-binding detection at the binary level. These metrics assess whether the model can identify pockets capable of lipid binding.

- **Micro ROC-AUC and Micro PR-AUC** are computed by flattening all pocket-category prediction pairs and treating each pair as an individual binary decision. Micro metrics summarize aggregated ranking performance across all lipid categories.

- **Per-category ROC-AUC and Per-category PR-AUC** are computed by evaluating each lipid category independently. They reveal category-specific behavior and reduce the risk that rare lipid classes are obscured by frequent ones.

Unless otherwise stated, AUC metrics are averaged over five random seeds and reported as mean $\pm$ standard deviation.

In addition, to quantify the model's ability to prioritize lipid-binding pockets in screening, we report:

- **Top-K precision (Hit@$K$).** We rank pockets by the predicted lipid-binding score in descending order and compute the fraction of positively annotated lipid-binding pockets among the top-$K$ retrieved pockets:

$$\text{Hit@}K = \frac{1}{K} \sum_{i=1}^{K} y_i,$$

  where $y_i \in \{0, 1\}$ denotes the lipid-binding annotation of the $i$-th pocket in the ranked list. Hit@$K$ is reported as the average over five runs with different random seeds.

## D. SLiPP Baseline Details

### D.1. Problem Setup and Pocket Representation

SLiPP (Chou et al., 2025; Chou & Dassama, 2024) is a pocket-level method for lipid-binding site identification. It constructs "lipid-binding" and "non-lipid-binding" pocket sets from ligand-bound protein structures in the Protein Data Bank (PDB) (Berman et al., 2000) by curating a restricted subset of ligand types. The lipid set includes cholesterol (CLR) and four fatty acids: myristic acid (MYR), palmitic acid (PLM), stearic acid (STE), and oleic acid (OLA). As emphasized in the original work (Chou et al., 2025), these ligands were selected primarily because each has sufficient resolved complexes to support model development. In contrast, several major lipid classes such as phospholipids, sphingolipids, and glycerolipids are excluded due to limited structural coverage. The non-lipid set is defined using a subset of representative metabolites and cofactors, including adenosine (ADN) for nucleosides, $\beta$-D-glucose (BGC) for saccharides, and cobalamin (B12) and coenzyme A (COA) for common cofactors. Consequently, SLiPP's training is tied to a small subset of ligand types, which may limit generalization to ligand classes outside the curated set.

*Table 5.* Statistics of the fpocket-derived easy/hard test sets.

| Split | Size | Labeled | FA | GL | GP | SP | ST | PR | SL | PK |
|---|---|---|---|---|---|---|---|---|---|---|
| easy-fpocket | 21,220 | 1,519 | 602 | 43 | 104 | 24 | 167 | 462 | 3 | 285 |
| hard-fpocket | 11,352 | 957 | 403 | 33 | 96 | 2 | 291 | 130 | 2 | 106 |

For each protein-ligand complex, SLiPP defines candidate pockets using the dpocket module from fpocket (Le Guilloux et al., 2009; Schmidtke et al., 2010). dpocket outputs ligand-overlapping cavities as well as additional non-overlapping cavities, which SLiPP refers to as pseudo-pockets. Lipid-binding examples are defined as ligand-overlapping pockets from lipid-bound complexes. Non-lipid-binding examples include pockets from non-lipid complexes and pseudo-pockets from lipid-bound complexes. Each candidate pocket is encoded using the 17 fpocket descriptors that summarize coarse geometric and physicochemical properties. SLiPP follows a fully supervised training setup. The prediction task is binary, classifying each pocket as lipid-binding or non-lipid-binding, and SLiPP does not produce lipid category-specific predictions.

For completeness, the 17 fpocket descriptors used by SLiPP are: pocket volume, pocket solvent-accessible surface area (SASA), pocket polar solvent-accessible surface area, pocket apolar solvent-accessible surface area, number of alpha spheres, mean alpha-sphere radius, mean alpha-sphere solvent accessibility, proportion of apolar alpha spheres, mean local hydrophobic density, hydrophobicity score, volume score, polarity score, charge score, proportion of polar atoms, alpha-sphere density, maximum distance from pocket center to alpha spheres, and druggability score.

### D.2. Training Protocol and fpocket-derived Evaluation Sets

SLiPP evaluated multiple classical machine-learning models and reported Random Forest as the best-performing option. We follow the original protocol and train SLiPP using its official open-source dataset with the Random Forest model (Chou et al., 2025; Chou & Dassama, 2024). We do not modify any design choices in SLiPP, including its curated ligand scope, label definitions, or pocket construction procedure. We repeat training for five independent runs with different random seeds.

To evaluate SLiPP under a consistent setting with our proposed LipoPU method, we construct fpocket-derived counterparts of our easy and hard test sets. The procedure is as follows. Starting from the annotated pockets in the BioLiP test splits, we run fpocket on the corresponding experimentally resolved PDB structures to generate candidate pockets for each protein. For each fpocket-predicted pocket with residue set $P_i$, we compute its maximum overlap with the experimental pocket residue sets $\{R_j\}$ of the same protein using an intersection-over-union (IoU) criterion:

$$s_i = \max_j \frac{|P_i \cap R_j|}{|P_i \cup R_j|}.$$

If $s_i \geq 0.4$, we assign $P_i$ the lipid-binding label of the experimental pocket $R_j$, which attains the maximum overlap. Predicted pockets with $s_i < 0.4$ are excluded from the evaluation. These selected results are most similar with experimental pockets, which helps ensure label validity. Collecting all such matched pockets yields the easy-fpocket and hard-fpocket evaluation sets. These sets preserve the original easy and hard splits while replacing experimental pockets with their best-matching fpocket-derived pockets.

SLiPP uses the 17 fpocket descriptors as its input pocket features for evaluation on these fpocket-based test sets. In contrast, LipoPU is evaluated on the same fpocket-derived pockets using residue-level ESM-2 representations computed over the residues in each predicted pocket, rather than using experimentally annotated pocket residues. Detailed test set sizes and per-lipid-category counts are reported in Table 5.

## E. Additional Evaluation Results

### E.1. Hard Test Results with Supervised Baselines

Figure 5 shows per-label PR-AUC and ROC-AUC on the hard test split for lipid category prediction, comparing LipoPU with supervised BCE baselines trained with and without the positive-guaranteed batch sampler. As expected, all methods degrade under remote homology, indicating that category-level specificity is substantially harder to predict than binary lipid-binding detection across unseen protein families, especially for rare categories. As noted in prior work, lipid classes can share similar physicochemical and structural pocket patterns, and the scarcity of class-specific positives further increases the difficulty in distinguishing them (Dong & Wu, 2025). Despite these challenges, LipoPU remains the strongest and most

stable method across most categories, supporting that PU learning better preserves category-specific signals than absolute negative labeling under incomplete supervision. The polyketide label highlights a distinct challenge. All methods perform poorly on polyketides on the hard split, even though it is not among the scarcest categories. One plausible explanation is that polyketides span an exceptionally diverse chemical space rather than a coherent family (Hertweck, 2009), so a single coarse label may be insufficient and finer-grained subclass labels could be required for reliable prediction.

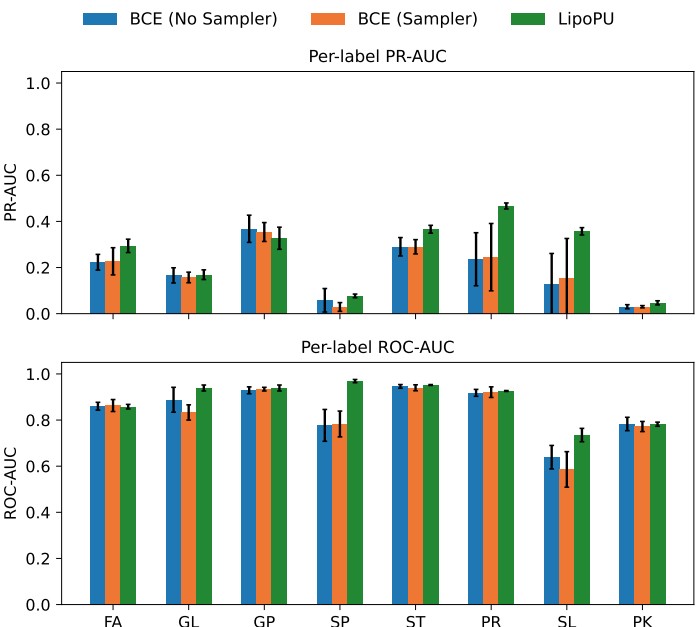

*Figure 5.* Per-label performance on the **Hard** test set. Bars report PR-AUC (top) and ROC-AUC (bottom) for each lipid category, comparing fully supervised BCE baselines (with/without our positive-guaranteed batch sampler) against LipoPU.

### E.2. Top-K Ranking Performance with Supervised Baselines

Figure 6a reports Hit@K for binary lipid-binding detection on the BioLiP2 easy and hard test splits ($K = 5, 10, 20, 50, 100$; mean over five runs), comparing LipoPU with supervised BCE baselines trained with and without the positive-guaranteed batch sampler. LipoPU consistently achieves the highest Hit@K on both splits and maintains near-saturated precision even at $K = 100$, indicating strong enrichment of annotated lipid-binding pockets among the top-ranked predictions. While the sampler improves BCE's Hit@K, BCE still depends on absolute negative labeling and therefore has limited ability to prioritize true lipid binders. In contrast, the ranking-based PU objective directly optimizes positive prioritization, leading to more confident retrieval of lipid-binding pockets.

### E.3. Top-K Ranking Performance with SLiPP

Figure 6b reports Hit@K on the easy-fpocket and hard-fpocket test splits ($K = 5, 10, 20, 50, 100$; mean over five runs), comparing LipoPU with SLiPP on fpocket-derived candidate pockets. LipoPU is comparable to SLiPP at small K and becomes clearly higher at larger cutoffs, while SLiPP declines sharply as K increases. Performance drops on the hard-fpocket split for both methods due to the combined difficulty of remote homology and noisy pocket boundaries, but LipoPU maintains higher Hit@K at larger K, consistent with better robustness under mismatched pocket definitions.

### E.4. Per-Category Results on fpocket-Derived Test Sets

Table 6 reports per-label PR-AUC and ROC-AUC for multi-label lipid category prediction on the easy-fpocket and hard-fpocket test splits. This evaluation is only applicable to LipoPU, since SLiPP does not support category-level prediction. LipoPU maintains competitive performance on the easy-fpocket set, capturing category signals despite mismatched pocket

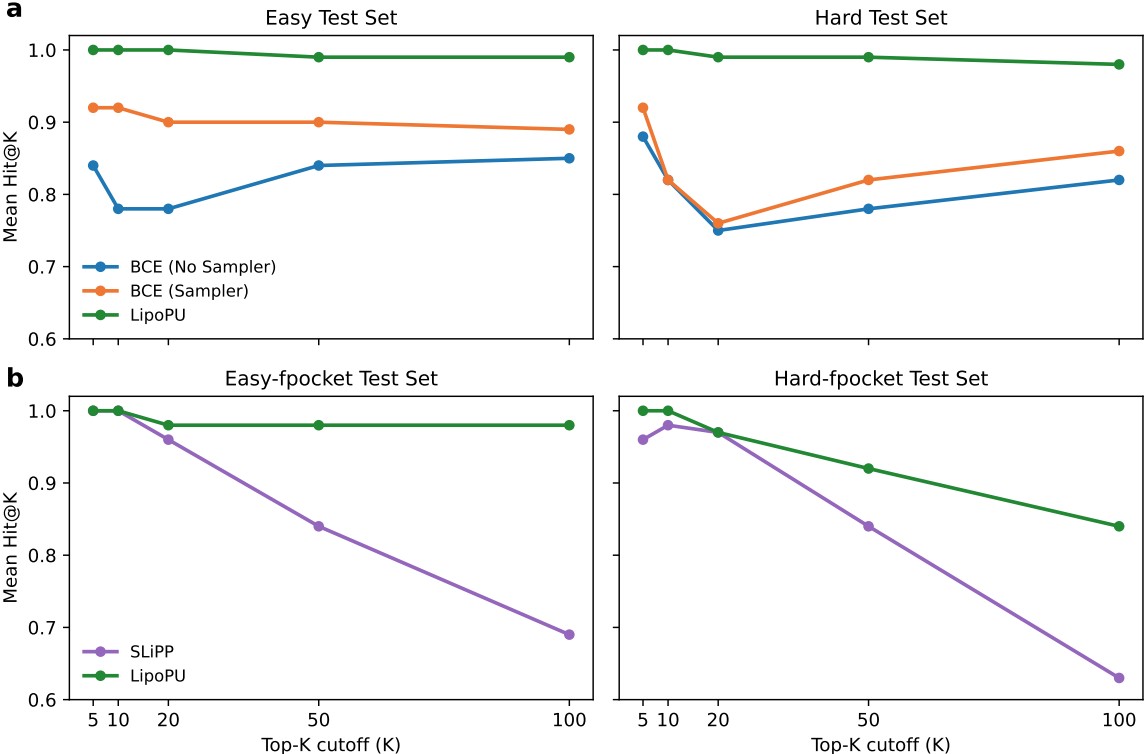

*Figure 6.* Top-$K$ ranking performance for binary lipid-binding detection. Hit@$K$ denotes the fraction of annotated lipid-binding pockets among the top-$K$ pockets ranked by the predicted lipid-binding score. Reported results are averaged over five runs with different random seeds. **a.** Results on BioLiP2 easy and hard test sets, comparing LipoPU with fully supervised baselines trained with binary cross-entropy, with and without the positive-guaranteed batch sampler. **b.** Results on fpocket-derived test sets (easy-fpocket and hard-fpocket), comparing LipoPU with SLiPP.

definitions. Performance drops on hard-fpocket, and PR-AUC declines much more than ROC-AUC. Consistent with the ranking degradation, these results suggest that reliably identifying true positives and predicting their lipid categories becomes particularly challenging under remote homology and pocket-definition shift.

### E.5. Per-Category Results for the Ablation Study

Figure 7 reports per-category PR-AUC and ROC-AUC for lipid category prediction on the BioLiP2 easy and hard test splits under our ablations. Removing the positive-guaranteed sampler reduces performance for most categories, showing that ensuring a minimum number of positives per batch is important for learning category-specific signals under extreme imbalance. Replacing the ranking-based PU objective with a standard PU formulation leads to the largest degradation, with PR-AUC dropping to very low values for several categories. This indicates that for the lipid binding prediction task, the ranking-based objective is essential for PU training to work reliably. Finally, mean pooling is often competitive in ROC-AUC but consistently worse in PR-AUC than attention-based MIL pooling across most classes, consistent with attention pooling better capturing category-discriminative signals. Overall, the per-category results highlight that the proposed sampler and ranking-based PU objective are critical to avoid category-wise collapse, while attention-based pooling provides a consistent advantage by emphasizing residues that are most informative for high-confidence retrieval.

### E.6. AT1R Case Study Category-Level Predictions

Following prior experimental setups (Lu et al., 2021; Zhang et al., 2023), we apply LipoPU to two pockets extracted from the AT1R cryo-EM structure, one from BioLiP2 annotations and the other from fpocket predictions. Table 7 reports the

*Table 6.* LipoPU performance on fpocket-derived test sets (easy vs. hard). Values are mean $\pm$ std over 5 runs.

| Split | Metric | FA | GL | GP | SP | ST | PR | SL | PK |
|---|---|---|---|---|---|---|---|---|---|
| easy-fpocket | PR-AUC | 0.692±0.008 | 0.849±0.020 | 0.824±0.011 | 0.695±0.027 | 0.769±0.016 | 0.865±0.004 | 0.785±0.091 | 0.498±0.051 |
| | ROC-AUC | 0.957±0.003 | 0.999±0.0003 | 0.990±0.002 | 0.9996±0.0002 | 0.991±0.001 | 0.989±0.002 | 1.000±0.00002 | 0.948±0.003 |
| hard-fpocket | PR-AUC | 0.211±0.028 | 0.230±0.046 | 0.209±0.051 | 0.501±0.0002 | 0.413±0.027 | 0.194±0.023 | 0.400±0.122 | 0.064±0.010 |
| | ROC-AUC | 0.816±0.010 | 0.988±0.005 | 0.884±0.024 | 0.957±0.008 | 0.924±0.003 | 0.880±0.009 | 0.718±0.073 | 0.757±0.011 |

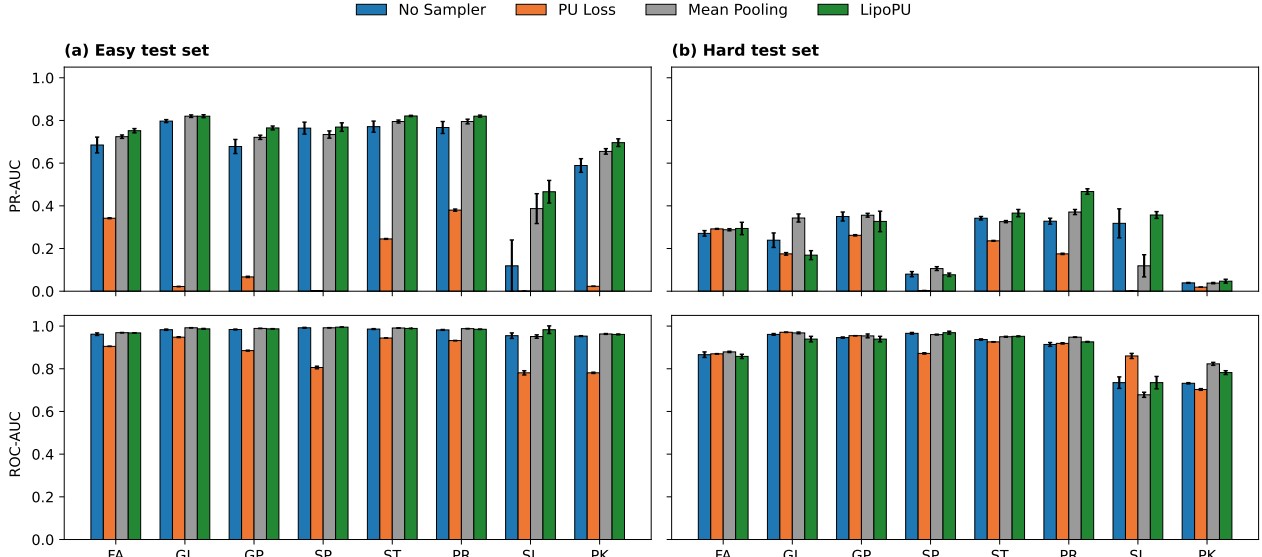

*Figure 7.* Ablation results for lipid category prediction on BioLiP2 test sets. Bars show per-category PR-AUC (top) and ROC-AUC (bottom), reported as mean $\pm$ standard deviation over five random seeds. Panel **a** reports results on the easy test set, and panel **b** reports results on the hard test set. We compare LipoPU with variants that remove the positive-guaranteed batch sampler (*No Sampler*), replace the ranking-based PU objective with a standard PU formulation (*PU Loss*), or replace attention-based pooling with mean pooling (*Mean Pooling*).

predicted scores over the eight lipid categories. For both pockets, the Sterol lipids (ST) category receives a near-unity score and is clearly separated from all other categories (0.999 for the BioLiP2 pocket and 0.987 for the fpocket-derived pocket), consistent with experimental reports of cholesterol binding at this site (Lu et al., 2021). The consistency between the two pockets extracted by different procedures indicates that LipoPU produces stable predictions under reasonable variations in pocket boundaries, supporting its practical use when pocket boundaries are uncertain.

### E.7. A stronger supervised alternative: Supervised Pairwise Ranking

To examine whether the improvement of LipoPU comes from ranking alone, rather than from the PU formulation, we implemented an additional supervised-style ranking baseline. This baseline replaces the pointwise weighted BCE loss with a pairwise logistic ranking loss, while treating unlabeled pockets as negatives under the conventional supervised assumption. Concretely, for each annotated positive pocket $x$, we sample an unlabeled pocket $y$ as the negative counterpart and optimize

$$\mathbb{E}_{x \sim P,\, y \sim U}\left[\ell(g(x) - g(y))\right].$$

Intuitively, this objective directly encourages annotated positives to receive higher scores than unlabeled pockets. For this new baseline, we keep the same architecture, class-prior weighting, and sampling strategy as in the main experiments. Together with our original baselines **Weighted BCE (sampler)** and **Standard PU (pointwise)**, the results are shown in Table 8.

Under this setting, LipoPU remains the strongest method. Importantly, the newly added Supervised Pairwise Ranking baseline performs even worse than the pointwise Weighted BCE (sampler), despite using an explicit ranking objective. This

*Table 7.* Per-category LipoPU prediction scores for the AT1R case study. Pocket residues are defined either by BioLiP2 experimental annotations or by fpocket predictions. Bold indicates the maximum score within each row.

| Pocket source | FA | GL | GP | SP | ST | PR | SL | PK |
|---|---|---|---|---|---|---|---|---|
| BioLiP2 | 0.024 | $7.1 \times 10^{-5}$ | $1.6 \times 10^{-4}$ | $4.5 \times 10^{-4}$ | **0.999** | $1.8 \times 10^{-4}$ | $3.0 \times 10^{-5}$ | $1.2 \times 10^{-4}$ |
| fpocket | 0.006 | $9.3 \times 10^{-5}$ | 0.007 | $4.7 \times 10^{-4}$ | **0.987** | $3.9 \times 10^{-4}$ | $5.9 \times 10^{-5}$ | $5.8 \times 10^{-5}$ |

*Table 8.* Comparison with a supervised-style pairwise ranking baseline. Values are mean $\pm$ std over 5 runs.

| Set | Method | Binary ROC-AUC | Binary PR-AUC | Micro ROC-AUC | Micro PR-AUC |
|---|---|---|---|---|---|
| Easy | *Supervised Pairwise Ranking* | 0.904±0.004 | 0.357±0.020 | 0.937±0.004 | 0.264±0.018 |
| | Weighted BCE (sampler) | 0.938±0.041 | 0.651±0.154 | 0.972±0.014 | 0.490±0.155 |
| | Standard PU (pointwise) | 0.899±0.001 | 0.449±0.002 | 0.929±0.0004 | 0.261±0.001 |
| | **LipoPU (ranking-based PU)** | **0.973±0.001** | **0.839±0.009** | **0.984±0.0004** | **0.766±0.010** |
| Hard | *Supervised Pairwise Ranking* | 0.889±0.005 | 0.363±0.015 | 0.849±0.017 | 0.130±0.009 |
| | Weighted BCE (sampler) | 0.904±0.019 | 0.558±0.069 | 0.898±0.014 | 0.199±0.033 |
| | Standard PU (pointwise) | 0.895±0.001 | 0.525±0.003 | **0.926±0.001** | 0.208±0.001 |
| | **LipoPU (ranking-based PU)** | **0.914±0.003** | **0.650±0.013** | 0.924±0.007 | **0.280±0.015** |

degradation is especially clear on the hard remote-homology split, where Binary PR-AUC drops from 0.558 with Weighted BCE to 0.363 with Supervised Pairwise Ranking, and Micro PR-AUC drops from 0.199 to 0.130. Relative to LipoPU, the drop is even larger, where Binary PR-AUC decreases from 0.650 to 0.363, and Micro PR-AUC decreases from 0.280 to 0.130.

This complementary test indicates that LipoPU's improvement does not come from ranking alone. Rather, the critical difference is whether the objective explicitly models annotation incompleteness. In supervised pairwise ranking, annotated positives are optimized to rank above unlabeled pockets that may contain latent positives, which can amplify the false-negative bias rather than alleviate it.

### E.8. An additional direct binary PU model

In LipoPU, lipid-binding detection is derived from the multi-label category scores by taking the maximum score across lipid categories, $S_{\text{bind}} = \max_l S_l$. This design reflects the definition of binary lipid binding, where a pocket is considered lipid-binding if it is associated with at least one lipid category. It also couples binary detection and lipid category prediction through a shared predictor, allowing the binary score to be informed by category-specific evidence.

To examine whether this max-over-categories design introduces harmful error propagation, we trained an additional direct binary PU model. This model is optimized only on the binary lipid-binding label, while keeping the same backbone, pocket representation, PU learning framework, and training pipeline as LipoPU. This provides a controlled comparison between the coupled max-over-categories design and a directly optimized binary PU objective. The results are shown in Table 9.

The direct binary PU objective yields similar binary performance overall and does not show a consistent advantage across splits. On the easy split, the max-over-categories design performs better, improving Binary ROC-AUC from 0.969 to 0.973 and Binary PR-AUC from 0.807 to 0.839. On the hard split, the direct binary PU objective is slightly higher, with Binary ROC-AUC increasing from 0.914 to 0.916 and Binary PR-AUC increasing from 0.650 to 0.676. These differences are modest, indicating that deriving binary lipid-binding scores from category-level predictions does not materially degrade binary prediction quality.

We further evaluate whether the max-over-categories design induces overconfident binary scores. Calibration is assessed under observed labels using Expected Calibration Error (ECE), which measures the mismatch between predicted confidence and the observed positive rate. The max-over-categories design achieves lower ECE than the direct binary PU objective on both splits, with 0.0037 versus 0.0206 on the easy split and 0.0154 versus 0.0312 on the hard split. We also compute the high-score false-positive rate at threshold 0.8, defined as the fraction of observed false positives among predictions with score at least 0.8. The two designs show similar high-score false-positive rates, with 0.0439 versus 0.0451 on the easy split and 0.0587 versus 0.0561 on the hard split. Thus, even when the binary score is derived by taking the maximum over eight category scores, we do not observe a clear increase in high-confidence observed false positives.

Taken together, under the observed labels, these results do not support the concern that max-over-categories systematically produces more overconfident binary scores or substantially more high-confidence false positives. Overall, the two formula-

*Table 9.* Comparison between the direct binary PU objective and the max-over-categories design used in LipoPU. Values are mean $\pm$ std over 5 runs.

| Set | Model | Binary ROC-AUC | Binary PR-AUC |
|---|---|---|---|
| Easy | Direct binary PU objective | $0.969\pm0.002$ | $0.807\pm0.009$ |
| | **LipoPU (max over categories)** | $\mathbf{0.973\pm0.001}$ | $\mathbf{0.839\pm0.009}$ |
| Hard | Direct binary PU objective | $\mathbf{0.916\pm0.005}$ | $\mathbf{0.676\pm0.007}$ |
| | **LipoPU (max over categories)** | $0.914\pm0.003$ | $0.650\pm0.013$ |

tions are competitive in binary ranking performance, while the max-over-categories design provides a natural way to couple binary detection with category-aware prediction in a unified predictor.

