# OpenReview forum: "LipoPU: Pocket-level Prediction of Lipid-Protein Interactions via Positive-Unlabeled Learning"
_ICML.cc/2026/Conference — ICML 2026 regular_

### Official Review · Reviewer_1wUt · 2026-03-07

**Soundness:** 2
**Presentation:** 2
**Significance:** 2
**Originality:** 1
**Overall Recommendation:** 3
**Confidence:** 4

**Summary:**

This paper introduces  a novel framework for pocket-level prediction of lipid-protein interactions.They used ranking-based Positive-Unlabeled (PU) learning to mitigate the systematic false negatives caused by incomplete experimental annotations. LipoPU utilizes an attention-based Multiple Instance Learning pooling mechanism to generate robust pocket representations from ESM-2 embeddings, enabling both binary lipid-binding detection and multi-label lipid category prediction. The method outperformed supervised baselines and the existing pocket-level method SLiPP, particularly in precision-recall metrics under severe class imbalance. Furthermore, the model provides residue-level interpretability, successfully identifying biologically relevant allosteric pockets in case studies.

**Compliance With Llm Reviewing Policy:**

Affirmed.

**Final Justification:**

While the authors has showed efforts, this topic has relatively limited interests. There are flaws in comparison with existing methods.

**Key Questions For Authors:**

see the above weakness.

**Limitations:**

yes

**Strengths And Weaknesses:**

Strengths
1. The shift from fully supervised learning to a ranking-based PU learning paradigm is well-motivated and addresses a critical flaw in current bioinformatics approaches where "absence of evidence" is incorrectly treated as " is particularly relevant for lipid-protein interactions where experimental data is sparse.
2. The method focuses on structural pockets, offering a more mechanistic understanding of binding sites and aligns better with the physical reality of lipid-protein interactions.
3. The experiments are rigorous, covering both binary detection and multi-label category prediction. The inclusion of a "hard test set" (remote homology) and comparisons against strong baselines (including SLiPP and supervised BCE variants) convincingly demonstrate the method's generalization capabilities.


Weaknesses
1. The ranking based strategy has been widely used in bioinformatics during ML era, although usually this brought incremental changes. As expected, this strategy brought little improvements.
2. Similarly, pocket-based strategy was commonly used, especially for docking studies. Though this limits searching range, the potential mis-prediction will cause following mistakes.
3. There are tons of studies in this field, but the authors didn't compare with SOTA methods.

---

> ### Author Rebuttal · Authors · 2026-03-31
>
> We thank the reviewer for the careful reading and detailed feedback. We appreciate the recognition that treating unlabeled data as negatives is a critical flaw, especially for lipid-protein binding prediction. We are also encouraged that the reviewer recognized the relevance of the pocket-level formulation and the evidence of generalization in our experiments.
>
> **Q1: The ranking-based strategy has been widely used in bioinformatics and brought only limited improvement.**
>
> We agree that ranking itself is not new. Our contribution is not ranking alone, but formulating **pocket-level lipid-binding prediction under incomplete annotation** as a **ranking-based PU problem**.
>
> To test whether the gain comes from ranking alone, we added a **stronger supervised pairwise ranking** control under the same backbone, priors, and sampling. This baseline forces annotated positives to rank above unlabeled samples treated as negatives.
>
> | Set | Method | Binary PR | Micro PR |
> | --- | --- | --- | --- |
> | Easy | Supervised pairwise ranking | 0.357 | 0.264 |
> | Easy | Weighted BCE | 0.651 | 0.490 |
> | Easy | **LipoPU** | **0.839** | **0.766** |
> | Hard | Supervised pairwise ranking | 0.363 | 0.130 |
> | Hard | Weighted BCE | 0.558 | 0.199 |
> | Hard | **LipoPU** | **0.650** | **0.280** |
>
> The results suggest that the gain is **not** from generic ranking alone. Under the wrong supervision assumption, pairwise ranking can even be harmful, because it explicitly pushes positives above an unlabeled pool that still contains latent positives. By contrast, **LipoPU** achieves clearly superior results by combining ranking with a PU formulation that models annotation incompleteness.
>
> **Q2: Pocket-based strategy is common, but pocket misprediction may propagate downstream errors.**
>
> Thanks for the question. We note that this concern involves two different uses of pockets. In docking pipelines, pockets are often used as search constraints. In our work, we address a different problem: pocket-level lipid-binding prediction given candidate pockets, rather than de novo pocket discovery or full docking.
>
> We agree that imperfect pocket definition is an important practical concern. We address this in two ways. First, beyond experimentally annotated pockets, we also evaluate on fpocket-derived candidate pockets that better reflect realistic conditions. LipoPU still clearly outperforms SLiPP, with PR-AUC improving from 0.239 to 0.792 on easy-fpocket and from 0.264 to 0.505 on hard-fpocket, showing that predictive utility remains despite pocket-definition noise. Second, our attention-based MIL pooling provides residue-level interpretability within each candidate pocket, highlighting which residues drive the prediction when boundaries are ambiguous.
>
> Our contribution is therefore not pocket use alone, but the combination of pocket-level representation with ranking-based PU learning for lipid-binding prediction under incomplete supervision.
>
> **Q3: There are many studies in this field, but the paper does not compare with SOTA methods.**
>
> We agree that broader comparison is valuable. This concern may partly reflect a mismatch between general ligand–protein prediction and the specific lipid-binding task studied here. There are many studies on general ligand–protein interactions. However, to the best of our knowledge, there is no directly matched prior method for **pocket-level, multi-label lipid-binding prediction under incomplete annotation**.
>
> To strengthen the comparison, we added two reproducible external baselines under a unified pocket-level evaluation.
>
> **DisoFLAG** is a residue-level predictor for lipid-binding-related functions in IDRs. We obtained residue scores from the official model, mapped them to pocket residues, and aggregated them by max, mean, and top3mean, with **top3mean** giving the best results.
>
> **PLiCat** is a sequence-level multi-label lipid-category predictor. We obtained the protein-level prediction vector from the official model and assigned the same vector to every pocket on that protein. This adaptation is reproducible, but it cannot localize which pocket drives the prediction.
>
> | Method | Adaptation | Easy ROC | Easy PR | Hard ROC | Hard PR |
> | --- | --- | --- | --- | --- | --- |
> | DisoFLAG | residue to pocket | 0.499 | 0.042 | 0.378 | 0.049 |
> | PLiCat | sequence to pocket | 0.340 | 0.031 | 0.304 | 0.045 |
> | **LipoPU** | pocket level | **0.973 ± 0.001** | **0.839 ± 0.009** | **0.914 ± 0.003** | **0.650 ± 0.013** |
>
> These results show that LipoPU performs best under a unified pocket-level evaluation. The added baselines are informative, but their native prediction granularity is mismatched to pocket discrimination. We therefore believe the key contribution is that LipoPU addresses a distinct setting: **pocket-level lipid-binding and category prediction under incomplete supervision**.
>
> We sincerely thank the reviewer again for the careful feedback and hope these additions help address the main concerns.

---

> > ### Author Rebuttal · Reviewer_1wUt · 2026-04-01
> >
> > The authors should test their methods on the DisoFLAG/PLiCat's datasets. The authors claims to be the first"pocket-level, multi-label lipid-binding prediction under incomplete annotation" method, but a method with such limits doesn't have enough wide interests. The authors need dig out why other methods has worse than random performance.

---

> > > ### Author Response · Authors · 2026-04-02
> > >
> > > We thank the reviewer for the additional feedback. We understand the concerns as threefold: whether the task is too narrow, why LipoPU was not evaluated on the original DisoFLAG/PLiCat datasets, and why the transferred baselines perform poorly. We address these points below.
> > >
> > > **Q1 (Scope).** We would like to clarify that “pocket-level, multi-label lipid-binding prediction under incomplete annotation” describes the method, not a narrowly limited problem setting. Our core contributions are three general methodological principles:
> > >
> > > 1. explicit modeling of incomplete annotation in binding prediction
> > > 2. alignment between the prediction unit and the biological decision unit
> > > 3. a rigorous multi-label PU formulation for diverse binding patterns under partial labels
> > >
> > > We focus on lipid binding because it is both biologically important and methodologically representative of these principles. In lipid-binding data, incomplete annotation is a major challenge due to systematic missing labels and strong context dependence, making it a particularly natural PU setting. Similar challenges also arise in broader domains such as protein–nucleic acid binding and allosteric site prediction.
> > >
> > > Furthermore, pockets naturally align with biological practice and should not be overlooked in binding prediction. Because binding is realized at specific local interaction sites, pocket-centered analysis and virtual screening are central to modern drug discovery. Pocket-level granularity is therefore a necessary problem formulation rather than a restriction of scope.
> > >
> > > **Q2 (Dataset)**. We agree that cross-dataset evaluation is generally valuable. Here, however, the issue is not merely dataset shift. In biology, sequence, residue, and pocket are distinct levels rather than interchangeable resolutions of the same object. They correspond to different units of molecular organization and biological mechanism.
> > >
> > > Accordingly, our adapted-baseline tests are designed to show that sequence- and residue-level supervision cannot replace native pocket-level modeling for pocket discrimination. Reversing that evaluation would therefore not provide a fairer test of the same task, but instead shift the biological unit and mechanism under study.
> > >
> > > Critically, the label structures are fundamentally misaligned across granularities. Mapping to residue-level settings (e.g., DisoFLAG) discards cross-residue cooperative signals and breaks the pocket as a geometric unit. Aggregating to the sequence level (e.g., PLiCat) collapses multiple candidate pockets into a single score, removing the ability to localize and rank binding sites. This is not an edge case but a general property of lipid-binding systems. For example, the ovine system b⁰,⁺ transporter contains two distinct lipid-binding pockets for LBN and CLR, corresponding to a phospholipid site and a conserved cholesterol site. Collapsing such a system to a single protein-level label erases both the spatial separation and the lipid-category distinction across different pockets.
> > >
> > > **Q3 (Baseline Analysis).** We thank the reviewer for requesting deeper analysis. This analysis is indeed informative.
> > >
> > > For **DisoFLAG**, the core issue is score compression and ranking failure. It is designed for lipid-binding residues within intrinsically disordered regions rather than coherent structural pockets. Across 24,272,231 predictions, the median score is 0.0023 and only 0.0056% exceed 0.5. More importantly, high-scoring pockets are not enriched for positives: on the easy split, the top 10% contain 2.84% positives versus 4.42% overall; on the hard split, 1.43% versus 6.56%. This reveals a substantial mismatch between IDR-derived residue signals and structured candidate-pocket targets.
> > >
> > > For **PLiCat**, broadcast adaptation cannot localize pocket-specific signal by construction. Its outputs are strongly biased toward positive predictions: even at the score threshold of 0.7, 94.58% remain positive. This is consistent with its original training distribution: PLiCat's training data contains 94.3% positives and only 5.7% negatives. Under standard supervised learning, such skewed priors naturally bias the model toward frequent classes and positive predictions, and may not match the class balance encountered beyond its original training setting.
> > >
> > > Taken together, these results show that the poor performance of the adapted baselines is structural rather than incidental, arising from mismatches in granularity, structural assumptions, and label distributions. More broadly, they show that pocket-level lipid-binding prediction is a genuinely challenging problem that demands a native formulation aligning the prediction unit, structural regime, and supervision assumptions. LipoPU is designed precisely for this setting.
> > >
> > > We again thank the reviewer for the feedback and hope these clarifications help resolve the main concerns.

---

### Official Review · Reviewer_nE9G · 2026-03-07

**Soundness:** 4
**Presentation:** 4
**Significance:** 3
**Originality:** 3
**Overall Recommendation:** 5
**Confidence:** 4

**Summary:**

This paper points out three issues encountered by existing methods of learning-based lipid-binding prediction: (1) potential false negatives, (2) suboptimal target representation, and (3) improper choice of prediction granularity.
To tackle these issues, this paper proposes LipoPU, a pocket-level framework that is charactrized by a pocket-level pooling method, a ranking-based objective, and a postive-guaranteed sampling strategy.
Experiments show that LipoPU outperforms supervised baselines and existing pocket-level methods.

**Compliance With Llm Reviewing Policy:**

Affirmed.

**Key Questions For Authors:**

See "Weaknesses"

**Limitations:**

Yes

**Strengths And Weaknesses:**

**Strengths**
* This paper identifies three important issues in existing methods. They are all insightful observations. In particular, the issue of potential false negative poses a challenge for ML methods, not only for lipid-binding prediction but also for other tasks related to biological data.
* The main design of LipoPU is interesting and make sense. Each component itself is not complex, but the components are combined together in a reasonable way.
* This paper is quite well written. The paragraphs are well organized and the figures are helpful. I really enjoyed reading this paper.
* The experimental results are good. The ablation studies support the design choices well.

**Weaknesses**
* It would be good to give some details about the notations and equations in Sec.3.3. For example, how are risks such as $\hat{R}_+(g)$ calculated? What does the equations mean intuitively?
* Technically, the LipoPU framework is not quite novel as each component has been well studied in prior works. Nevertheless, I do not believe this should lead to a rejection decision.

---

> ### Author Rebuttal · Authors · 2026-03-31
>
> We sincerely thank the reviewer for the highly positive and insightful comments. We are especially encouraged that you recognized the false-negative issue as important not only for lipid-binding prediction, but also more broadly for machine learning on biological data. We also deeply appreciate your positive assessment of the design, presentation, and empirical results of our work.
>
> **Q1: It would be good to give some details about the notations and equations in Sec.3.3. For example, how are risks such as $\hat{R}\_{+}(g)$ calculated? What do the equations mean intuitively?**
>
> We thank the reviewer for this helpful suggestion. We agree that Sec. 3.3 should explain both the intuition and the empirical computation more explicitly, and we will revise it accordingly.
>
> Eqs. (4) to (6) describe the transition from supervised risk to PU risk. In supervised learning, the risk is decomposed into positive and negative terms. In PU learning, however, negatives are not directly observed, and the unlabeled set \$U\$ is a mixture of true negatives and latent positives. The negative term is therefore rewritten by subtracting the positive contamination term, which is the key correction for false-negative bias.
>
> For each lipid category $l$, Eq. (7) defines the non-negative PU risk
> $$
> \hat{R}\_l(g)=\pi\_l\hat{R}\_{P,l}^{+}(g)+\max(0,\hat{R}\_{U,l}^{-}(g)-\pi\_l\hat{R}\_{P,l}^{-}(g)).
> $$ Here, $P\_l$ and $U\_l$ are the observed positive and unlabeled sets for label $l$, and $\pi\_l$ is the class prior estimated from the observed training frequency.
>
> Eqs. (8) to (10) are empirical loss terms written in expectation form. Intuitively, $\hat{R}\_{P,l}^{+}(g)$ encourages observed positives to receive high scores, $\hat{R}\_{P,l}^{-}(g)$ measures the penalty if positives were treated as negatives, and $\hat{R}\_{U,l}^{-}(g)$ is the unlabeled-as-negative term. In implementation, these expectations are estimated by mini-batch averages. For example
> $$
> \hat{R}\_{P,l}^{+}(g)=\frac{1}{|P\_l^{(b)}|}\sum\_{x\in P\_l^{(b)}}-\log\sigma(g\_l(x)),\quad
> \hat{R}\_{P,l}^{-}(g)=\frac{1}{|P\_l^{(b)}|}\sum\_{x\in P\_l^{(b)}}-\log\sigma(-g\_l(x)).
> $$
>
> Eq. (11) further replaces the unlabeled-negative term with a ranking surrogate
> $$
> \hat{R}\_{U,l}^{\mathrm{rank}}(g)=\mathbb{E}\_{x\sim P\_l,y\sim U\_l}\big[-\log\sigma(g\_l(x)-g\_l(y))\big].
> $$
> Its key intuition is that the model does not force all unlabeled pockets to have low absolute scores. Instead, it encourages observed positives to rank above unlabeled pockets on average, which is more appropriate when unlabeled data may contain latent positives. In practice, this is implemented as a batch-wise approximation using logits and positive and unlabeled masks.
>
> Thus, in our implementation, Eq. (7) becomes
> $$
> \hat{R}\_l(g)=\pi\_l\hat{R}\_{P,l}^{+}(g)+\max(0,\hat{R}\_{U,l}^{\mathrm{rank}}(g)-\pi\_l\hat{R}\_{P,l}^{-}(g)),
> $$
> and the final objective in Eq. (12) is
> $$
> \mathcal{L}(g)=\sum\_{l=1}^{L}\hat{R}\_l(g).
> $$
> We will incorporate these notation clarifications, mini-batch estimates, and intuitive explanations directly into Sec. 3.3 in the revision.
>
> **Q2: Technically, the LipoPU framework is not quite novel as each component has been well studied in prior works. Nevertheless, I do not believe this should lead to a rejection decision.**
>
> We thank the reviewer for this fair and constructive assessment. We agree that the individual components are not claimed as standalone algorithmic novelties. Rather, the contribution of this work lies in a rigorous and task-driven integration for pocket-level lipid-binding prediction under incomplete annotation, which, to our knowledge, is not explicitly addressed by prior methods.
>
> Rather than being an arbitrary combination of existing modules, each component is introduced to address a specific challenge in this problem setting. Ranking-based PU learning addresses false negatives caused by incomplete annotation. Pocket-level modeling targets the actual functional unit of lipid recognition. Attention-based MIL provides a robust and interpretable pocket representation under uncertain pocket boundaries. Positive-guaranteed sampling improves optimization stability under severe class imbalance and supports informative positive-versus-unlabeled comparisons during training.
>
> The contribution is therefore methodological and empirical, rather than a purely conceptual combination. Our ablations support the role of these design choices, and the full framework consistently improves over supervised baselines and prior pocket-level methods. We will clarify more explicitly in the revision that the paper’s contribution is this rigorous task-driven integration rather than entirely new standalone modules.
>
> We sincerely thank the reviewer again for the encouraging and constructive assessment. These comments help us substantially improve the clarity and positioning of the paper.

---

> > ### Author Rebuttal · Reviewer_nE9G · 2026-04-02
> >
> > Thank the authors for the response. My concerns are addressed and I will keep my rating.

---

> > > ### Author Response · Authors · 2026-04-02
> > >
> > > We sincerely thank you for your careful reading and thoughtful engagement throughout the discussion. We deeply appreciate the time, effort, and constructive feedback you provided, which helped us clarify the paper more effectively. It is encouraging to know that our response could address your concerns, and we are grateful for your positive assessment of our work.

---

### Official Review · Reviewer_izK2 · 2026-03-10

**Soundness:** 2
**Presentation:** 3
**Significance:** 2
**Originality:** 3
**Overall Recommendation:** 4
**Confidence:** 3

**Summary:**

This paper introduces LipoPU, a new lipid binding prediction model that tackles two major flaws in previous lipid-binding research: the biased practice of treating unlabeled samples as definitive negatives and the lack of focus on pocket level predictions compared to broader sequence based methods. This method uses a pocket centric approach and employing Positive Unlabeled (PU) learning with a ranking-based objective. It functions as both lipid-binding detection and lipid category prediction. LipoPU outperforms standard BCE-based models(treating unlabeled data as negative) and existing pocket-based tools like SLiPP.

**Compliance With Llm Reviewing Policy:**

Affirmed.

**Final Justification:**

the rebuttal addressed my major concern on limited baselines. Although I still think the scope of this paper is narrow.

**Key Questions For Authors:**

Please refer to weaknessess

**Limitations:**

There is no limitations and potential negative societal impact  section.

**Strengths And Weaknesses:**

## Strengths
1. The problem identified by this paper in the lipid binding prediction task is reasonable: treating unlabeled data as negative. In general, the use of positive–unlabeled learning in this area is rational. From the ablation results, the improvement brought by the PU loss is also quite clear.

2. The overall writing of this paper is good. The method is clearly described, and the research question is clearly stated. The figures, tables, and algorithms are also quite comprehensive. The appendix also contains extensive analyses.

## Weaknesses
1. The baseline in this paper is very limited. Comparing only with BCE makes it closer to an ablation study.
2. The domain this paper focuses on is somewhat too narrow. Why focus only on lipid binding rather than general protein–ligand binding?
3. Similarly, why focus only on the pocket level? Non–pocket-level approaches could also be compared.
4. The attention-based MIL pooling does not show a clear advantage over mean pooling. The technical contribution in this part is quite limited.

---

> ### Author Rebuttal · Authors · 2026-03-31
>
> We sincerely thank the reviewer for the careful and constructive feedback. We are encouraged that the reviewer recognizes false negative labeling as a genuine challenge in lipid binding prediction and agrees that PU learning is well motivated. To address the remaining concerns, we added stronger baselines, broader cross-granularity comparisons, and clearer discussion of scope and limitations. We hope these additions help support a more favorable reassessment.
>
> **Q1: Baselines are too limited**
>
> We agree that a BCE-only comparison may appear closer to an ablation study. While our original paper already included the prior pocket-level method SLiPP together with standard PU under the same backbone, we now broaden the comparison in two stronger directions.
>
> First, we added **a stronger supervised pairwise ranking model**, using the same architecture, sampler, and class prior setting. It enforces annotated positives to rank above "negative" pockets under the supervised assumption. Methods and results are reported in Reviewer 1wUt, Q1. LipoPU remains best on both splits, and supervised pairwise ranking is even worse than weighted BCE, especially on the hard split where binary PR AUC drops from 0.558 to 0.363 and micro PR AUC drops from 0.199 to 0.130. This supports our main claim that the improvement does not come from ranking alone, because ranking under an inappropriate supervision assumption can amplify false negative bias rather than correct it.
>
> Second, we added external baselines from other granularities under transparent adaptation to the same pocket benchmark. **DisoFLAG** is a residue-level predictor for lipid-related IDR functions. We obtain residue scores and aggregate pocket residues by max, mean, and top3mean, with top3mean performing best. **PLiCat** is a sequence-level lipid category predictor. We run the official model and assign the resulting sequence prediction to every pocket on that protein. Methods and results are reported in Reviewer 1wUt, Q3. Under this unified pocket evaluation, both adapted baselines are far below LipoPU. This goes beyond a BCE-only ablation. It covers prior pocket-level work, a stronger supervised ranking baseline, and external residue-level and sequence-level predictors.
>
> **Q2: Why lipid binding rather than general protein ligand binding**
>
> While lipid binding is a more focused setting, we believe it is well motivated and biologically meaningful. Lipid–protein interactions are biologically important and actively studied, with new binders being increasingly identified, suggesting incomplete annotations and making this a natural PU setting. In addition, lipids have distinct structural and physicochemical properties and are often tightly linked to hydrophobic or membrane context. A generic protein–ligand formulation would mix ligand families with different chemistries and annotation semantics. We therefore focus on a coherent and underexplored setting where PU learning is especially well justified, while broader ligand classes remain important future work.
>
> **Q3: Why pocket-level rather than other granularities**
>
> We focus on the pocket-level because it provides the structural granularity that best matches the question of where lipid binding occurs. Compared with residue-level prediction, it captures the coordinated physicochemical environment formed by multiple residues. Compared with protein-level prediction, it can localize binding sites rather than assigning a single score to the entire protein. To further address this concern, **we include comparisons with non–pocket-level methods (DisoFLAG and PLiCat)**. Their limited performance under the same pocket-level benchmark suggests a granularity mismatch, supporting the need for a native pocket-level formulation.
>
> **Q4: MIL pooling shows limited gain**
>
> We agree and will revise the wording accordingly to avoid overstating this part. We do not claim that attention-based MIL is the main source of the empirical gain. Mean pooling is a strong baseline. We use MIL because candidate pocket boundaries are often ambiguous and only a subset of residues may be truly informative. Its main value is residue weighting and interpretability within a pocket, not a large standalone performance jump.
>
> **Q5: Limitations and potential negative impact**
>
> We agree that the limitations in the Conclusion and Impact Statement should be made more explicit. In the revision, we will clarify two further points. First, the current results support ranking and candidate prioritization, not biological confirmation of binding. Second, the annotation incompleteness motivating PU learning also affects training and evaluation, since observed labels may be biased across lipid types. Reported metrics should therefore be interpreted primarily as evidence of prioritization utility under incomplete annotation.
>
> We thank the reviewer again for the feedback and hope the added clarifications help address the concerns.

---

> > ### Author Rebuttal · Reviewer_izK2 · 2026-04-03
> >
> > The main technical concern (limited baselines) has been addressed. While the scope remains somewhat narrow and the analysis could be deeper, the paper is now a solid contribution with clearer empirical support. I will increase the score.

---

> > > ### Author Response · Authors · 2026-04-07
> > >
> > > We sincerely thank the reviewer for the thoughtful follow-up and for the generous reassessment of our work. We are very encouraged that the empirical support and overall contribution of the paper are now clearer. This is genuinely motivating for our further work.
> > >
> > > We also understand your remaining concerns, particularly regarding the scope and the depth of analysis. We hope the following clarification may be helpful.
> > >
> > > **Q1 (Scope).** Although this task is focused on lipid binding, its broader contribution lies in three general methodological principles:
> > >
> > > 1. explicit modeling of incomplete annotation in binding prediction
> > > 2. alignment between the prediction unit and the biological decision unit
> > > 3. a rigorous multi-label PU formulation for diverse binding patterns under partial labels
> > >
> > > We focus on lipid binding because it provides a representative instance of these principles. Lipid-binding data are strongly affected by missing labels and context dependence, making them a natural PU problem. Similar challenges also arise in broader domains such as protein–nucleic acid binding and allosteric site prediction. At the same time, binding is realized at localized structural sites, so pocket-level modeling is not an arbitrary restriction, but an indispensable unit that naturally aligns with biological practice.
> > >
> > > **Q2 (Deeper analysis).** We agree that deeper analysis is valuable, and our further analysis of the adapted baselines helps clarify the cross-granularity differences.
> > >
> > > For **DisoFLAG**, the core issue is score compression and ranking failure. It is designed for lipid-binding residues in intrinsically disordered regions rather than coherent structural pockets. Across 24,272,231 residues, the median score is 0.0023 and only 0.0056% exceed 0.5. More importantly, high-scoring pockets are not enriched for positives: in the top 10%, positives are 2.84% vs. 4.42% overall on the easy split, and 1.43% vs. 6.56% on the hard split. This indicates a substantial mismatch between IDR-derived residue signals and structured candidate-pocket targets.
> > >
> > > For **PLiCat**, broadcast adaptation cannot localize pocket-specific signal by construction. Its outputs are strongly biased toward positive predictions: even at the score threshold of 0.7, 94.58% remain positive. This is consistent with its original training distribution: PLiCat's training data contains 94.3% positives and only 5.7% negatives. Under standard supervised learning, such skewed priors naturally bias the model toward frequent classes and positive predictions, and may not match the class balance encountered beyond the original training setting.
> > >
> > > Taken together, these results suggest that the limitations of the adapted baselines are not incidental, but arise from mismatches in granularity and structural assumptions. Residue-level transfer discards cross-residue cooperative signals and fails to preserve the pocket as a geometric unit. Sequence-level aggregation collapses multiple candidate pockets into a single score, removing the ability to localize and rank binding sites. This further supports the need for a native pocket-level formulation.
> > >
> > > We again sincerely thank the reviewer for the careful reading and constructive feedback throughout the discussion. We hope this clarification is helpful in addressing the remaining concerns.

---

### Official Review · Reviewer_UJDc · 2026-03-10

**Soundness:** 3
**Presentation:** 3
**Significance:** 2
**Originality:** 3
**Overall Recommendation:** 4
**Confidence:** 3

**Summary:**

This paper studies pocket-level prediction of lipid–protein interactions under incomplete labels. The authors claim to analyze a pertinent question: whether experimentally unlabeled pockets should be treated as negatives, or instead modeled with positive-unlabeled learning because annotation absence may reflect missing evidence rather than non-binding. The method, LipoPU, combines frozen ESM-2 residue embeddings, attention-based MIL pooling for pocket representation, a ranking-based multi-label PU objective, and a positive-guaranteed batch sampler. The paper evaluates both binary lipid-binding detection and multi-label lipid category prediction on a large curated dataset of 395,959 pockets with 18,402 positives, using easy and remote-homology hard splits. Reported results show improvements over supervised BCE baselines and over the prior pocket-level method SLiPP, plus an AT1R case study illustrating residue-level attention patterns.

**Compliance With Llm Reviewing Policy:**

Affirmed.

**Final Justification:**

The rebuttal addressed my main concerns, reinforcing my prior assessment.

**Key Questions For Authors:**

What exactly is the batch construction policy in the multi-label setting? Algorithm 1 describes positive-guaranteed batching at the instance level, while Appendix B states that the PU ranking loss enforces at least 1 positive per label in each mini-batch. Please clarify whether the sampler guarantees positives globally, per label, or both, and how this is implemented when rare labels co-occur sparsely. This would affect my view of both correctness and reproducibility.


Did the authors compare the current max-over-categories binary head to a directly supervised binary PU objective? A convincing answer could clarify whether the coupling between tasks is an advantage or a potential source of error propagation.



Can the authors clarify the strongest baseline set they considered and why certain plausible baselines were omitted? In particular, did they try stronger PU baselines, sequence-level category predictors adapted to pocket evaluation, or more expressive pocket representations without the PU ranking loss? This would help separate gains due to representation, objective, and training protocol, and could change my assessment of originality and empirical completeness.

**Limitations:**

The authors do discuss limitations and potential negative impact to a meaningful extent: the conclusion acknowledges degradation under pocket-definition shifts, data scarcity for several lipid types, and coarse label granularity, while the Impact Statement notes possible over-trust in predictions and bias toward lipids that are easier to resolve experimentally (Conclusion; Impact Statement). That said, I would encourage the authors to make two additions. First, explicitly distinguish “ranking for prioritization” from “binding confirmation,” since the current evidence mainly supports ranking quality rather than calibrated biological confirmation. Second, discuss more directly how incomplete positive labels may affect both training and evaluation, since the same annotation incompleteness that motivates PU learning can also blur benchmark labels.

**Strengths And Weaknesses:**

- Strengths


The paper clearly motivates why lipid-binding prediction is a natural PU setting: unlabeled pockets are not guaranteed negatives, and the introduction gives a concrete GPR119 example to support this claim. The manuscript also explains why pocket-level prediction is mechanistically relevant relative to sequence- or residue-level alternatives, and why lipid category specificity matters beyond binary detection. This framing is coherent and biologically meaningful.

The architecture is easy to follow: frozen ESM-2 embeddings, pocket residue selection, attention-based MIL pooling, category-wise MLP outputs, and binary detection obtained by max over category scores. The role of each component is stated explicitly, and the design aligns with the stated goals of ambiguity robustness and residue-level interpretability.

The split design uses MMseqs2 clustering at 30% identity and defines both easy and hard test sets, with the hard split sharing less than 30% identity with other splits. This is an appropriate choice for a biosequence/structure setting where memorization across homologs is a concern.



In Table 1, LipoPU improves over BCE variants on both easy and hard sets for binary and micro metrics, with especially large gains in PR-AUC. This is important because PR-AUC is more informative under the severe class imbalance described in Sec. 4.1.

Table 3 shows clear drops when replacing the ranking-based PU loss with a standard PU loss, and smaller but noticeable drops when removing the sampler, especially in PR-AUC. This gives reasonably direct evidence that the main empirical gains are not solely from the encoder/classifier backbone.


- Weaknesses




Fairness of the SLiPP comparison is improved but still not entirely symmetric. The paper uses the same fpocket-derived test pockets for both methods, but SLiPP uses handcrafted fpocket descriptors while LipoPU uses ESM-2 residue embeddings from the fpocket residues (Sec. 4.3). This is a reasonable “best use of each method” comparison, yet it is not a same-feature comparison, so the source of gain is a mixture of learning paradigm, representation quality, and possibly capacity.


The binary prediction is derived as max over category scores, which may create calibration and dependency issues that are not analyzed. Sec. 3.1 and Fig. 1 define binary lipid-binding detection as S_{bind}=\max_l S_l. This is simple, but the manuscript does not analyze whether this induces overconfident binary scores when any category fires spuriously, nor whether direct binary supervision/objectives were compared.


Per-label class imbalance is extreme for some categories, but the consequences are only partially explored. Table 4 shows very small counts for some labels, especially SL, and the conclusion notes limited positives for several lipid types. While Fig. 2 and Appendix references suggest per-label analysis, the main paper does not deeply discuss which categories remain unreliable or how uncertainty differs by label.

---

> ### Author Rebuttal · Authors · 2026-03-31
>
> We thank the reviewer for the careful reading and positive assessment. We are encouraged that the reviewer recognizes the motivation for PU learning and the value of pocket-level, category-aware modeling.
>
> **Q1. SLiPP comparison is not fully symmetric.**
>
> We agree that this is a best-use comparison against the prior pocket-level method rather than a same-feature controlled attribution study. In such method-level comparisons, differences in representation, objective, and capacity are inherently coupled. To reduce asymmetry, both methods are evaluated on the same fpocket-derived easy/hard test sets. We therefore interpret this result as external method-level evidence under realistic scenarios, while attribution of gains is isolated by our internal ablations in Tables 1 and 3.
>
> **Q2/Q5. Does max-over-categories harm binary prediction or calibration relative to a direct binary PU objective?**
>
> **We added a controlled comparison by training a direct binary PU model with the same backbone and pipeline.** Results are very similar: Easy ROC-AUC/PR-AUC 0.969/0.807 (direct binary PU) vs 0.973/0.839 (max); Hard 0.916/0.676 vs 0.914/0.650. Thus, **max-over-categories does not show a consistent disadvantage**.
>
> We also checked **calibration** under observed labels. First, Expected Calibration Error (ECE) measures the mismatch between predicted confidence and the observed positive rate. Max-over-categories has lower ECE than direct binary PU (Easy 0.0037 vs 0.0206, Hard 0.0154 vs 0.0312), indicating that its confidence is **better** aligned with the observed labels. Second, High-FP rate@0.8 measures the fraction of observed false positives among predictions with score ≥ 0.8. The results are close (Easy 0.0439 vs 0.0451, Hard 0.0587 vs 0.0561). Thus, **max-over-categories does not show systematic overconfidence or harmful error propagation, while naturally coupling binary and category prediction.**
>
> **Q3. Extreme per-label imbalance and label-specific uncertainty.**
>
> We agree this should be discussed more explicitly. Rare labels such as SL and SP have extremely sparse positives, which makes both learning and evaluation less stable. This is reflected in the high-confidence region. On the easy split, the top-100 predictions are dominated by FA and PR, while SL and SP do not appear; on the hard split, the top-100 is dominated by ST, and GL/GP/SP/SL/PK do not appear. We will state more clearly that results for rare categories, especially SL and SP, should be interpreted with additional caution.
>
> **Q4. What does the sampler guarantee?**
>
> Our implementation is instance-level global positive-guaranteed, not per-label guaranteed. The Appendix sentence is a wording error, and we will correct it. When a rare label is absent from a batch, no label-specific positive is injected. Supervision for that label is therefore accumulated across the subset of batches. The sampler is designed to avoid fully unlabeled batches and stabilize optimization, not to distort the natural long-tail distribution of rare categories.
>
> **Q6. What is the strongest baseline set, and why were some baselines omitted?**
>
> Our strongest baseline is not a single model but a **factorized comparison matrix**:
>
> (1) **Objective** under the same backbone, pooling, and sampler: Weighted BCE, standard PU, ranking-based PU (LipoPU), and **supervised pairwise BCE** added in rebuttal.
>
> (2) **Training protocol** under the same model and loss: with or without the positive-guaranteed sampler.
>
> (3) **Task granularity** on the same pocket-level benchmark: SLiPP, plus **PLiCat and DisoFLAG** added in rebuttal.
>
> (4) **Representation** under the same objective and protocol: mean pooling vs. attention-based MIL.
>
> In rebuttal, we added a stronger supervised pairwise BCE baseline and two cross-granularity baselines, PLiCat and DisoFLAG, adapted to the same pocket-level splits. Methods and results are in our response to Reviewer 1wUt, Q1 and Q3. Both remain far below LipoPU after adaptations. We also examined iDLB-Pred and DisoLipPred, but publicly available resources do not support reproducible large-scale inference in our setting, and both target IDR residues rather than structured pockets.
>
> Overall, the evidence supports LipoPU as a carefully controlled design that fills a real gap: pocket-level lipid-binding prediction with lipid-category specificity under incomplete supervision.
>
> **Q7. Clarify limitations and scope.**
>
> We agree and will make two points more explicit. First, our results support ranking and candidate prioritization, not biological confirmation of binding. Second, the same annotation incompleteness that motivates PU learning also affects both training and evaluation, as observed labels may be biased across lipid types. Reported metrics should therefore be interpreted primarily as evidence of prioritization utility under incomplete annotation.
>
> We thank the reviewer again. These comments helped us sharpen the methodological clarification and the scope of our claims.

---

> > ### Author Rebuttal · Reviewer_UJDc · 2026-04-02
> >
> > Thank you. My concerns are resolved.

---

> > > ### Author Response · Authors · 2026-04-02
> > >
> > > We sincerely thank you for your acknowledgement and for the careful reading of our work throughout the discussion. We are deeply grateful for the time, effort, and thoughtful feedback you provided, which helped us improve both the clarification and the positioning of the paper. It is very encouraging to know that our response could address your concerns.

---

### Decision · Program_Chairs · 2026-04-30

**Decision:**

Accept (regular)

**Comment:**

This paper proposes LipoPU, a well-motivated pocket-level framework for lipid-protein interaction prediction that effectively uses positive-unlabeled (PU) learning to address missing experimental annotations. The reviewers appreciated the sound PU formulation, clear architecture, and great empirical improvements over standard baselines.

While initial concerns were raised regarding the narrow biological scope and limited baselines, the authors provided a comprehensive rebuttal with additional baseline comparisons (e.g., PLiCat, DisoFLAG) that resolved most issues. Although one reviewer maintained a weak reject due to scope concerns, the committee consensus firmly leans positive. The method offers a solid, task-driven contribution to computational biology.